# Communication-efficient Algorithms Under Generalized Smoothness Assumptions

## Abstract

We provide the first proof of convergence for normalized error feedback algorithms across a wide range of machine learning problems. Despite their popularity and efficiency in training deep neural networks, traditional analyses of error feedback algorithms rely on the smoothness assumption that does not capture the properties of objective functions in these problems. Rather, these problems have recently been shown to satisfy generalized smoothness assumptions, and the theoretical understanding of error feedback algorithms under these assumptions remains largely unexplored. Moreover, to the best of our knowledge, all existing analyses under generalized smoothness either i) focus on single-node settings or ii) make unrealistically strong assumptions for distributed settings, such as requiring data heterogeneity, and almost surely bounded stochastic gradient noise variance. In this paper, we propose distributed error feedback algorithms that utilize normalization to achieve the $\mathcal{O}(1/\sqrt{K})$ convergence rate for nonconvex problems under generalized smoothness. Our analyses apply for distributed settings without data heterogeneity conditions, and enable stepsize tuning that is independent of problem parameters. Additionally, we provide strong convergence guarantees of normalized error feedback algorithms for stochastic settings. Finally, we show that normalized EF21, due to its larger allowable stepsizes, outperforms EF21 on various tasks, including the minimization of polynomial functions, logistic regression, and ResNet-20 training.

## 1 Introduction

Machine learning models achieve impressive prediction and classification power by employing sophisticated architectures, comprising vast numbers of model parameters, and requiring training on massive datasets. Distributed training has emerged as an important approach, where multiple machines with their own local training data collaborate to train a model efficiently within a reasonable time. Many optimization algorithms can be easily adapted for distributed training frameworks. For example, stochastic gradient descent (SGD) can be modified into distributed stochastic gradient descent within a data parallelism framework, and into federated averaging algorithms (McMahan et al., 2017) in a federated learning framework. However, the communication overhead of running these distributed algorithms poses a significant barrier to scaling up to large models. For example, training the VGG-16 model (Simonyan & Zisserman, 2015) using distributed stochastic gradient descent involves communicating 138.34 million parameters, thus consuming over 500MB of storage and posing an unmanageable burden on the communication network between machines.

One approach to mitigate the communication burden is to apply compression. In this approach, the information, such as gradients or model parameters, is compressed using sparsifiers or quantizers to be transmitted with much lower communicated bits between machines. However, while this reduces communication overhead, too coarse compression often brings substantial challenges in maintaining high training performance due to information loss, and in extreme cases, it may potentially lead to divergence. Therefore, error feedback mechanisms have been developed to improve the convergence performance of compression algorithms, while ensuring high communication efficiency. Examples of error feedback mechanisms include EF14 (Seide et al., 2014; Stich et al., 2018; Alistarh et al., 2018; Wu et al., 2018; Gorbunov et al., 2020), EF21 (Richtárik et al., 2021; Fatkhullin et al., 2021), EF21-SGDM (Fatkhullin et al., 2024), EF21-P (Gruntkowska et al., 2023), and EControl (Gao et al.,

2023). Several studies developing error feedback algorithms often assume the smoothness of an objective function, i.e., its gradient is Lipschitz continuous.

However, many modern learning problems, such as distributionally robust optimization (Jin et al., 2021) and deep neural network training, are often non-smooth. For instance, the gradient of the loss computed for deep neural networks, such as LSTM (Zhang et al., 2020b), ResNet20 (Zhang et al., 2020b), and transformer models (Crawshaw et al., 2022), is not Lipschitz continuous. These empirical findings highlight the need for a new smoothness assumption. One such assumption is $(L_0, L_1)$-smoothness, originally introduced by Zhang et al. (2020b), for twice differentiable functions, and later extended to differentiable functions by Chen et al. (2023).

To solve generalized smooth problems, clipping and normalization have been widely utilized in first-order algorithms. Gradient descent with gradient clipping was initially shown by Zhang et al. (2020b) to achieve lower iteration complexity, i.e., fewer iterations needed to attain a target solution accuracy, than classical gradient descent. Subsequent works have further refined the convergence theory of clipped gradient descent (Koloskova et al., 2023), and improved its convergence performance by employing momentum updates (Zhang et al., 2020a), variance reduction techniques (Reisizadeh et al., 2023), and adaptive step sizes (Wang et al., 2024; Li et al., 2024b; Takezawa et al., 2024). Similar convergence results have been obtained for gradient descent using normalization (Zhao et al., 2021), and its momentum variants (Hübler et al., 2024), including generalized SignSGD (Crawshaw et al., 2022). However, these first-order algorithms have mostly been explored in training on a single machine. To the best of our knowledge, distributed algorithms under generalized smoothness have been investigated in only a few works, e.g., by Crawshaw et al. (2024); Liu et al. (2022). Nonetheless, these works rely on assumptions limiting families of optimization problems, including data heterogeneity, almost sure variance bounds, and symmetric noise distributions around the mean assumptions. Furthermore, these first-order algorithms under generalized smoothness do not incorporate compression techniques to improve communication efficiency. These aspects motivate us to develop *distributed communication-efficient algorithms for solving nonconvex generalized smooth problems*.

## 1.1 CONTRIBUTIONS

In this paper, we develop distributed error feedback algorithms for communication-efficient optimization under nonconvex, generalized smooth regimes. Our contributions are summarized below.

● **Importance of normalization.** Just as gradient clipping is crucial for gradient descent, we empirically demonstrate that normalization stabilizes the convergence of error feedback algorithms for minimizing nonconvex generalized smooth functions. In this paper, we introduce a variant of EF21, a widely used error feedback algorithm by Richtárik et al. (2021), which incorporates normalization to guarantee convergence for nonconvex, generalized smooth problems. In a single-node setting, normalized EF21 provides larger stepsize, and faster convergence rate than original EF21 for minimizing simple nonconvex polynomial functions that satisfy generalized smoothness, as shown by Figure 1.

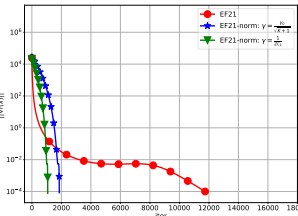 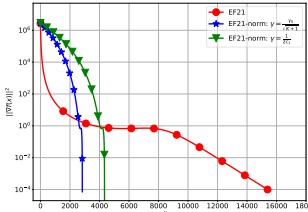 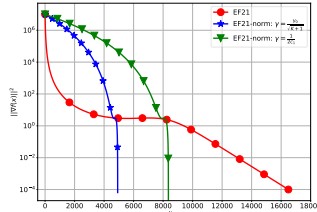

Figure 1: The minimization of polynomial functions using EF21 with $\gamma = \frac{1}{L + L\sqrt{\frac{\beta}{\theta}}}$, and normalized EF21 (`EF21-norm`) with $\gamma = \frac{\gamma_0}{\sqrt{K+1}}$, $\gamma_0 = 1$ (blue line) and $\gamma = \frac{1}{2c_1}$ (green line). Here, we ran both algorithms for (1) $L_0 = 4$, $L_1 = 1$, and $K = 2000$ (left), (2) $L_0 = 4$, $L_1 = 4$, and $K = 5000$ (middle), and (3) $L_0 = 4$, $L_1 = 8$, and $K = 16000$ (right).

- **Convergence of normalized error feedback algorithms.** We establish an $\mathcal{O}(1/\sqrt{K})$ convergence rate in the gradient norm for normalized EF21 on nonconvex generalized smooth problems. Normalized EF21 achieves the same rate as the original EF21 under $L$-smoothness by Richtárik et al. (2021). Our results are derived under standard assumptions, i.e., generalized smoothness and the existence of lower bounds on the objective function, and are applicable in distributed settings regardless of any data heterogeneity degree, unlike the results by Crawshaw et al. (2024); Liu et al. (2022). Additionally, our stepsize rules for normalized EF21 ensure convergence without requiring knowledge of the generalized smoothness constants $L_0$ or $L_1$, in contrast to Richtárik et al. (2021), where the stepsize depends on the smoothness constant $L$ (which is often inaccessible).

- **Extension to stochastic settings.** Furthermore, we propose a variant of EF21-SGDM, an error feedback algorithm with momentum updates by Fatkhullin et al. (2024), that employs normalization for solving nonconvex, stochastic optimization under generalized smoothness. Specifically, we prove that normalized EF21-SGDM with suitable stepsize choices attains the same $\mathcal{O}(1/K^{1/4})$ convergence rate in the gradient norm as the original EF21-SGDM.

- **Numerical evaluation.** We implemented normalized EF21 using the stepsize rules derived from our theory, and compared its performance against the original EF21. Both algorithms were evaluated on three learning tasks: minimizing nonconvex polynomial functions, solving logistic regression with a nonconvex regularizer, and training ResNet-20 on the CIFAR-10 dataset. Thanks to its larger stepsizes, normalized EF21 outperforms the original EF21, in terms of both convergence speed and solution accuracy across these tasks.

| Methods | Complexity | Smoothness | Variance bound | Normalization |
|---|---|---|---|---|
| EF21 Richtárik et al. (2021) | $\mathcal{O}(1/\epsilon^2)$ | $L$ | No | No |
| EF21-SGDM Fatkhullin et al. (2024) | $\mathcal{O}(1/\epsilon^4)$ | $L$ | expectation | No |
| Normalized EF21 Ours (Alg. 1) | $\mathcal{O}(1/\epsilon^2)$ | $(L_0, L_1)$ | No | Yes |
| Normalized EF21-SGDM Ours (Alg. 2) | $\mathcal{O}(1/\epsilon^4)$ | $(L_0, L_1)$ | Expectation | Yes |

Table 1: Comparisons of complexities and assumptions between known and our results for EF21 variants. The complexity is defined by the iteration count $K$ required by the algorithms to attain $\min_{k=0,1,\ldots,K} \mathrm{E}\left[\left\|\nabla f(x^k)\right\|\right] \leq \epsilon$. $(L_0, L_1)$-smoothness refers to generalized smoothness in Assumption 3. The variance bound in expectation is defined in Assumption 5.

## 2 RELATED WORKS

**Error feedback.** Error feedback mechanisms have been utilized in various algorithms with communication compression, leading to significant improvements in solution accuracy, while reducing communication. As the first version of these mechanisms, EF14 was introduced by Seide et al. (2014), and later analyzed for first-order algorithms in both single-node (Stich et al., 2018; Karimireddy et al., 2019) and distributed settings (Alistarh et al., 2018; Wu et al., 2018; Tang et al., 2019; Basu et al., 2019; Gorbunov et al., 2020; Li et al., 2020; Qian et al., 2021; Tang et al., 2021). Next, EF21 is another error feedback variant proposed by Richtárik et al. (2021), which offers strong convergence guarantees for distributed gradient algorithms with any contractive compressors, without requiring bounded gradient norm or bounded data heterogeneity assumptions. EF21 can also be adapted for stochastic optimization through sufficiently large mini-batches (Fatkhullin et al., 2021) or momentum updates (Fatkhullin et al., 2024). More recently, EControl was developed by Gao et al. (2023) to guarantee provably superior complexity results for distributed stochastic optimization compared to prior error feedback mechanisms. To the best of our knowledge, these existing works on error feedback have focused solely on optimization under traditional $L$-smoothness. In this paper, we introduce a normalized variant of the EF21 methods (Richtárik et al., 2021) for solving nonconvex generalized smooth problems. In particular, we prove that normalized EF21 under generalized

smoothness achieves the same $\mathcal{O}(1/\sqrt{K})$ rate as original EF21 under traditional smoothness, and demonstrate in experiments that normalized EF21 permits larger step sizes, and thus attains faster convergence than the original EF21.

**Non-smoothness assumptions.**  Empirical findings suggest that the traditional smoothness used for analyzing optimization algorithms does not capture the properties of objective functions in many machine learning problems, especially deep neural network training problems. This motivates researchers to consider different assumptions to replace this traditional smoothness condition. First introduced by Zhang et al. (2020b), the $(L_0, L_1)$-smoothness condition on a twice differentiable function $f(x)$ is defined by $\left\|\nabla^2 f(x)\right\| \leq L_0 + L_1 \left\|\nabla f(x)\right\|$ for $x \in \mathbb{R}^d$. This $(L_0, L_1)$-smoothness has been extended to differentiable functions without assuming the existence of the Hessian. For instance, the smoothness with a differentiable function $\ell(x)$ (Li et al., 2024a), and symmetric generalized smoothness (Chen et al., 2023) cover the $(L_0, L_1)$-smoothness when the Hessian exists, and includes many important machine learning problems, such as phase retrieval problems (Chen et al., 2023), and distributionally robust optimization (Levy et al., 2020). Other classes of non-smoothness assumptions, which are not related to the generalized smoothness but capture other optimization problems, include Hölder's continuity of the gradient (Devolder et al., 2014), the relative smoothness (Bauschke et al., 2017), and the polynomial growth of the gradient norm (Mai & Johansson, 2021). In this paper, we impose the generalized smoothness condition to establish the convergence of normalized EF21 for solving deterministic and stochastic optimization.

**Gradient clipping and normalization.**  Clipping and normalization are commonly employed in gradient-based methods for solving generalized smooth problems. Clipped (stochastic) gradient descent has been studied for both nonconvex and convex problems under $(L_0, L_1)$-smoothness conditions by Zhang et al. (2020b); Koloskova et al. (2023). Extensions to clipped gradient algorithms have been proposed, including momentum updates (Zhang et al., 2020a), variance reduction methods (Reisizadeh et al., 2023), and adaptive step sizes (Wang et al., 2024; Li et al., 2024b; Takezawa et al., 2024). Comparable complexities have been achieved for normalized gradient descent (Zhao et al., 2021), and its momentum-based variants (Hübler et al., 2024), including generalized SignSGD (Crawshaw et al., 2022). Convergence properties of gradient-based algorithms have also been explored under more generalized forms of non-uniform smoothness, extending beyond the $(L_0, L_1)$-smoothness by Zhang et al. (2020b) to cover a wider range of optimization problems. For example, variants of (stochastic) gradient descent have been analyzed under $\alpha$-symmetric generalized smoothness by Chen et al. (2023), and under $\ell$-smoothness involving certain differentiable functions $\ell(\cdot)$ by Li et al. (2024a;b). However, the majority of these analyses focus on the single-node setting. To the best of our knowledge, only a limited number of works, such as those by Crawshaw et al. (2024); Liu et al. (2022), have examined federated averaging algorithms for nonconvex problems under generalized smoothness. These works, however, often rely on restrictive assumptions, including data heterogeneity, almost sure variance bounds, and symmetric noise distributions centered around their means. In this paper, we develop distributed error feedback algorithms, which eliminate the need for the restrictive assumptions mentioned above, and rely on standard assumptions on objective functions and compressors.

## 3 PRELIMINARIES

**Notations.**  We use $[n]$ to denote the set $\{1, 2, \ldots, n\}$, and $\mathrm{E}[u]$ to represent the expectation of a random variable $u$. Additionally, $\|\cdot\|$ indicates the Euclidean norm for vectors or the spectral norm for matrices, and $\|\cdot\|_1$ is the $\ell_1$-norm for vectors, while $\langle x, y \rangle$ denotes the inner product between $x$ and $y$ in $\mathbb{R}^d$. Lastly, for a square matrix $A \in \mathbb{R}^{d \times d}$, $\lambda_{\min}(A)$ refers to its minimum eigenvalue, and $I \in \mathbb{R}^{d \times d}$ is the identity matrix.

**Problem formulation.**  In this paper, we focus on the following distributed optimization problem:

$$\min_{x \in \mathbb{R}^d} f(x) := \frac{1}{n} \sum_{i=1}^{n} f_i(x), \tag{1}$$

where $n$ refers to the number of clients, and $f_i(x)$ is the loss of a model parameterized by vector $x \in \mathbb{R}^d$ over its local data $\mathcal{D}_i$ owned by client $i \in [n]$.

**Assumptions.** To facilitate our convergence analysis, we make standard assumptions on objective functions and compression operators.

**Assumption 1.** *(Lower Bound of $f$) A function $f(x) = (1/n) \sum_{i=1}^n f_i(x)$ is bounded from below, i.e., $f^{\inf} = \inf_{x \in \mathbb{R}^d} f(x) > -\infty$.*

**Assumption 2.** *(Lower Bound of $f_i$) A function $f_i(x)$ is bounded from below, i.e., $f_i^{\inf} = \inf_{x \in \mathbb{R}^d} f_i(x) > -\infty$.*

Assumptions 1 and 2 are standard for analyzing optimization algorithms for unconstrained optimization.

**Assumption 3.** *(Generalized Smoothness of $f_i$) A function $f_i(x)$ is symmetrically generalized smooth if there exists $L_0, L_1 > 0$ such that for $u_\theta = \theta x + (1 - \theta) y$, and for all $x, y \in \mathbb{R}^d$*

$$\|\nabla f_i(x) - \nabla f_i(y)\| \le \left(L_0 + L_1 \sup_{\theta \in [0,1]} \|\nabla f_i(u_\theta)\|\right) \|x - y\|. \tag{2}$$

Assumption 3 refers to symmetric generalized smoothness defined by Chen et al. (2023), which covers asymmetric generalized smoothness (Koloskova et al., 2023; Chen et al., 2023), and the original $(L_0, L_1)$-smoothness by Zhang et al. (2020b). Moreover, Assumption 3 covers the functions with unbounded classical smoothness constant, e.g., exponential function. Additionally, Assumption 3 with $L_1 = 0$ reduces to the traditional $L_0$-smoothness (Nesterov et al., 2018; Beck, 2017), under which the convergence of optimization algorithms has been extensively studied.

**Assumption 4.** *(Contractive Compressor) An operator $\mathcal{C}^k : \mathbb{R}^d \to \mathbb{R}^d$ is an $\alpha$-contractive compressor if there exists $\alpha \in (0, 1]$ such that for $k \ge 0$ and $v \in \mathbb{R}^d$,*

$$\mathrm{E}\left[\left\|\mathcal{C}^k(v) - v\right\|^2\right] \le (1 - \alpha) \|v\|^2. \tag{3}$$

Furthermore, compressors defined by Assumption 4 cover top-$k$ sparsifiers (Alistarh et al., 2018; Stich et al., 2018), low-rank approximation (Vogels et al., 2019; Safaryan et al., 2021), and various other compressors described by Beznosikov et al. (2023); Safaryan et al. (2022).

**Assumption 5.** *(Bounded Variance) A stochastic gradient $\nabla f_i(x; \xi_i)$ with its sample $\xi_i \sim \mathcal{D}_i$ is an unbiased estimator of $\nabla f_i(x)$ with bounded variance, i.e.,*

$$\mathrm{E}\left[\nabla f_i(x; \xi_i)\right] = \nabla f_i(x), \quad and \quad \mathrm{E}\left[\|\nabla f_i(x; \xi_i) - \nabla f_i(x)\|^2\right] \le \sigma^2. \tag{4}$$

Assumption 5 is standard assumption for stochastic optimization (Nemirovski et al., 2009; Ghadimi & Lan, 2012; 2013) that is only imposed on each local stochastic gradient, and it does not imply data heterogeneity, i.e., the bounded difference between each component function $f_i(x)$ and the global function $f(x)$.

---

**Algorithm 1** Normalized EF21

---

1: **Input:** Stepsize $\gamma_k > 0$ for $k = 0, 1, \ldots$; starting points $x^0, v_i^{-1} \in \mathbb{R}^d$ for $i \in \{1, 2, \ldots, n\}$; and $\alpha$-contractive compressors $\mathcal{C}^k : \mathbb{R}^d \to \mathbb{R}^d$ for $k = 0, 1, \ldots$.
2: **for** each iteration $k = 0, 1, \ldots, K$ **do**
3:    **for** each client $i = 1, 2, \ldots, n$ in parallel **do**
4:       Compute local gradient $\nabla f_i(x^k)$
5:       Transmit $\Delta_i^k = \mathcal{C}^k(\nabla f_i(x^k) - v_i^{k-1})$
6:       Update $v_i^k = v_i^{k-1} + \Delta_i^k$
7:    **end for**
8:    Central server computes $v^k = \frac{1}{n} \sum_{i=1}^n v_i^k$ via $v_i^k = v_i^{k-1} + \Delta_i^k$
9:    Central server updates $x^{k+1} = x^k - \gamma_k \frac{v^k}{\|v^k\|}$
10: **end for**
11: **Output:** $x^{K+1}$

---

## 4 NORMALIZED EF21

For nonconvex deterministic optimization under generalized smoothness, we develop a distributed error feedback algorithm. One challenge is that the generalized smoothness parameter scales with the gradient norm $\left\|\nabla f(x^k)\right\|$. To resolve this issue, we apply gradient normalization to the algorithms. In particular, we consider normalized EF21, the normalized version of EF21 (Richtárik et al., 2021) that updates the next iterates $x^{k+1}$ using the normalized EF21 update. The full description of normalized EF21 can be found in Algorithm 1.

Normalized EF21, like EF21 (Richtárik et al., 2021) under traditional smoothness, enjoys the $\mathcal{O}(1/\sqrt{K})$ convergence in the gradient norm under generalized smoothness, as shown below.

**Theorem 1.** *Consider Problem (1), where Assumption 1 (lower bound of $f$), Assumption 2 (lower bound of $f_i$), Assumption 3 (generalized smoothness of $f_i$), and Assumption 4 (contractive compressor) hold. Then, the iterates $\{x^k\}$ generated by normalized EF21 (Algorithm 1) with*

$$\gamma_k = \frac{\gamma_0}{\sqrt{K+1}}$$

*for $K \geq 0$ and $\gamma_0 > 0$ satify*

$$\min_{k=0,1,\ldots,K}\mathrm{E}\left[\left\|\nabla f(x^k)\right\|\right] \leq \frac{V^0 \exp(8c_1 L_1 \exp(L_1\gamma_0)\gamma_0^2)}{\gamma_0\sqrt{K+1}} + B\frac{\gamma_0\exp(L_1\gamma_0)}{\sqrt{K+1}},$$

*where $V^k := f(x^k)-f^{\mathrm{inf}}+\frac{2\gamma_k}{1-\sqrt{1-\alpha}}\frac{1}{n}\sum_{i=1}^n\left\|\nabla f_i(x^k)-v_i^k\right\|$, $B = 2c_0+\frac{8L_1c_1}{n}\sum_{i=1}^n(f^{\mathrm{inf}}-f_i^{\mathrm{inf}})$, and $c_i = \left(\frac{1}{2}+2\frac{\sqrt{1-\alpha}}{1-\sqrt{1-\alpha}}\right)L_i$ for $i = 0, 1$.*

Theorem 1 establishes the $\mathcal{O}(1/\sqrt{K})$ convergence in the expectation of gradient norms for normalized EF1 on nonconvex deterministic problems under generalized smoothness. This rate is the same as Theorem 1 of Richtárik et al. (2021) for EF21 under traditional smoothness, and does not depend on data heterogeneity conditions in contrast to Crawshaw et al. (2024); Liu et al. (2022). Also, our stepsize depends on any positive constant $\gamma_0$, and total iteration count $K$, without needing to know smoothness constants $L_0, L_1$ in contrast to Richtárik et al. (2021). Additionally, if we choose $\gamma_0 = 1/(8cL_1)$, then our convergence bound from Theorem 1 becomes

$$\min_{k=0,1,\ldots,K}\mathrm{E}\left[\left\|\nabla f(x^k)\right\|\right] \leq \frac{32cL_1V^0 + L_0/L_1 + 2L_1\delta^{\mathrm{inf}}}{\sqrt{K+1}},$$

where $c = \frac{1}{2}+2\frac{\sqrt{1-\alpha}}{1-\sqrt{1-\alpha}}$, and $\delta^{\mathrm{inf}} = \frac{1}{n}\sum_{i=1}^n(f^{\mathrm{inf}}-f_i^{\mathrm{inf}})$.

**Comparisons between normalized EF21 and EF21 under traditional smoothness.** For nonconvex, traditional smooth problems, normalized EF21 from Theorem 1 with $L_1 = 0$ achieves the same $\mathcal{O}(1/\sqrt{K})$ rate in the expectation of gradient norms as EF21 analyzed by Richtárik et al. (2021), but with a larger convergence factor. We prove this by assuming $\nabla f_i(x^0) = v_i^0$ for all $i$. That is, Theorem 1 with $L_0 = L$, $L_1 = 0$, $\gamma_0 = \sqrt{(f(x^0)-f^{\mathrm{inf}})/(2b)}$, and $b = \frac{L}{2}+2\frac{\sqrt{1-\alpha}L}{1-\sqrt{1-\alpha}}$ implies that normalized EF21 achieves

$$\begin{aligned}
\min_{k=0,1,\ldots,K}\mathrm{E}\left[\left\|\nabla f(x^k)\right\|\right] &\leq \frac{1}{\sqrt{K+1}}\left[\frac{f(x^0)-f^{\mathrm{inf}}}{\gamma_0}+2b\gamma_0\right] \\
&\leq 2\sqrt{L\frac{(1+3\sqrt{1-\alpha})(1+\sqrt{1-\alpha})}{\alpha}}\sqrt{\frac{f(x^0)-f^{\mathrm{inf}}}{K+1}} \\
&\stackrel{\alpha\geq 0}{\leq} 4\sqrt{2}\sqrt{\frac{L}{\alpha}}\sqrt{\frac{f(x^0)-f^{\mathrm{inf}}}{K+1}}.
\end{aligned}$$

On the other hand, EF21 attains from Theorem 1 of Richtárik et al. (2021) with $L_i = \tilde{L} = L$ (i.e., $f_i(x)$ has the same smoothness constant as $f(x)$), and $\hat{x}^K$ being chosen from the iterates

$x^0, x^1, \ldots, x^K$ uniformly at random

$$
\begin{aligned}
\min_{k=0,1,\ldots,K} \mathrm{E}\left[\|\nabla f(x^k)\|\right] &\leq \mathrm{E}\left[\|\nabla f(\hat{x}^K)\|\right] \\
&\leq \sqrt{\mathrm{E}\left[\|\nabla f(\hat{x}^K)\|^2\right]} \\
&\leq \sqrt{2L(1 + \sqrt{\beta/\theta})\frac{f(x^0) - f^{\inf}}{K+1}} \\
\overset{\sqrt{\beta/\theta} \leq 2/\alpha - 1}{\leq} \quad & 2\sqrt{\frac{L}{\alpha}}\sqrt{\frac{f(x^0) - f^{\inf}}{K+1}}.
\end{aligned}
$$

In conclusion, the convergence bound of normalized EF21 is slower by a factor of $2\sqrt{2}$ than the original EF21 for nonconvex, $L$-smooth problems.

While normalized EF21 can handle nonconvex problems under generalized smoothness, the algorithm is limited to deterministic settings, where each node computes its full local gradient. In the following section, we demonstrate how to integrate normalization into EF21-SGDM Fatkhullin et al. (2024), an error feedback algorithm that allows each node to compute its local stochastic gradient, for solving nonconvex stochastic problems.

---

**Algorithm 2** Normalized EF21-SGDM

---

1: **Input:** Stepsizes $\gamma_k > 0$ and $\eta_k \in [0,1]$ for $k = 0, 1, \ldots$; starting points $x^0, g_i^{-1} \in \mathbb{R}^d$ for $i \in \{1, 2, \ldots, n\}$, and $v_i^0 = \frac{1}{B^{\init}}\sum_{j=1}^{B^{\init}} \nabla f_i(x_i^0; \xi_{i,j}^0)$ with i.i.d. random samples $\xi_{i,j}$ for $i \in \{1, 2, \ldots, n\}$ and an initial mini-batch size $B^{\init}$; $\alpha$-contractive compressors $\mathcal{C}^k : \mathbb{R}^d \to \mathbb{R}^d$ for $k = 0, 1, \ldots$
2: **for** each iteration $k = 0, 1, \ldots, K$ **do**
3:     **for** each client $i = 1, 2, \ldots, n$ in parallel **do**
4:         Compute a local stochastic gradient $\nabla f_i(x^k; \xi_i^k)$
5:         Update a momentum estimator $v_i^k = (1 - \eta_k)v_i^{k-1} + \eta_k \nabla f_i(x^k; \xi_i^k)$
6:         Transmit $\Delta_i^k = \mathcal{C}^k(v_i^k - g_i^{k-1})$
7:         Update $g_i^k = g_i^{k-1} + \Delta_i^k$
8:     **end for**
9:     Central server computes $g^k = (1/n)\sum_{i=1}^n g_i^k$ via $g_i^k = g_i^{k-1} + \Delta_i^k$
10:     Central server updates $x^{k+1} = x^k - \gamma_k \frac{g^k}{\|g^k\|}$
11: **end for**
12: **Output:** $x^{K+1}$

---

## 5 NORMALIZED EF21-SGDM

Having established the convergence of normalized EF21 for deterministic optimization, we will next develop a distributed error feedback algorithm that incorporate stochastic gradients and normalization to accommodate generalized smoothness conditions. In particular, we focus on normalized EF21-SGDM (Algorithm 2), the normalized version of EF21-SGDM (Fatkhullin et al., 2024). We also note that normalized EF21-SGDM recovers many optimization algorithms of interest in the special cases. For instance, normalized EF21-SGDM reduces to normalized EF21 when we let $\eta_k = 1$ and $\nabla f_i(x^k; \xi_i^k) = \nabla f_i(x^k)$, the normalized version of EF21-SGD (Fatkhullin et al., 2021) when we let $\eta_k = 1$, and normalized SGD with momentum (Cutkosky & Mehta, 2020) (NSGD-M) when we let $\eta_k = 1 - \beta_k$ and $\mathcal{C}^k(\cdot) \equiv I$.

In the next theorem, we demonstrate that normalized EF21-SGDM attains the same $\mathcal{O}(1/K^{1/4})$ convergence rate as both EF21-SGDM and NSGD-M.

**Theorem 2.** *Consider Problem (1), where Assumption 1 (lower bound of $f$), Assumption 2 (lower bound of $f_i$), Assumption 3 (generalized smoothness of $f_i$), Assumption 4 (contractive compressor), and Assumption 5 (bounded variance) hold. If the mini-batch size at the starting point $B^{\text{init}} \equiv \sqrt{K+1}$, and the stepsizes*

$$\gamma_k \equiv \gamma = \frac{\gamma_0}{(K+1)^{3/4}}, \text{ with } \gamma_0 > 0 \text{ satisfying } \gamma_0 \exp(\gamma_0 L_1/2) \leq \frac{1}{8L_1\sqrt{1+\sqrt{1-\alpha}/\alpha}}, \quad \text{and}$$

$$\eta_k \equiv \eta = \frac{1}{(K+1)^{1/2}},$$

*then the iterates $\{x^k\}$ generated by normalized EF21-SGDM (Algorithm 2) satisfy for $K \geq 0$*

$$\min_{k=0,1,\ldots,K} \mathrm{E}\left[\left\|\nabla f(x^k)\right\|\right] \leq \mathcal{O}\left(\frac{\mathrm{E}[V^0]/\gamma_0 + \sigma/\sqrt{n} + (\gamma_0 L_0 + \gamma_0 L_1^2 \delta^{\text{inf}}) \exp(\gamma_0 L_1)}{(K+1)^{1/4}}\right)$$

$$+ \mathcal{O}\left(\frac{\sqrt{1-\alpha}}{\alpha} \cdot \frac{\sigma + (L_0\gamma_0 + \gamma_0 L_1^2 \delta^{\text{inf}}) \exp(\gamma_0 L_1)}{(K+1)^{1/2}}\right),$$

*where $V^0 = f(x^0) - f^{\text{inf}} + \frac{2\gamma}{(1-\sqrt{1-\alpha})n}\sum_{i=1}^n \left\|v_i^0 - g_i^0\right\|$, and $\delta^{\text{inf}} = \frac{1}{n}\sum_{i=1}^n (f^{\text{inf}} - f_i^{\text{inf}})$.*

From Theorem 2, normalized EF21-SGDM under generalized smoothness achieves the $\mathcal{O}(1/K^{1/4})$ convergence rate in the expectation of gradient norms. This rate is the same as that of EF21-SGDM, previously analyzed under traditional smoothness by Fatkhullin et al. (2024, Theorem 3). The result holds regardless of the data heterogeneity degree and the mini-batch size, with the exception that the mini-batch size at the initial point (when $k = 0$) must satisfy $B_{\text{init}} = \sqrt{K+1}$ for a fixed $K \geq 0$. Additionally, one possible for the stepsize $\gamma_0 > 0$ satisfying the condition from Theorem 2 is $\gamma_0 \leq 1/(9L_1\sqrt{1+B(\alpha)})$ with $B(\alpha) = \sqrt{1-\alpha}/\alpha$. Notice that the stepsize $\gamma_0$ for normalized EF21-SGDM, unlike in the case of normalized EF21, depends on the generalized smoothness constant $L_1$, and the compression parameter $\alpha$.

Furthermore, Theorem 2 with $\alpha = 1$ (i.e., $\mathcal{C}^k(\cdot) \equiv I$) implies the convergence bound of the distributed version of normalized SGD with momentum (NSGD-M) (Cutkosky & Mehta, 2020) using $\beta = 1 - \eta$:

$$\min_{k=0,1,\ldots,K} \mathrm{E}\left[\left\|\nabla f(x^k)\right\|\right] \leq \mathcal{O}\left(\frac{(f(x^0)-f^{\text{inf}})/\gamma_0 + \sigma/\sqrt{n} + \gamma_0 L_0 + \gamma_0 L_1^2 \delta^{\text{inf}}}{(K+1)^{1/4}}\right). \quad (5)$$

For the single-node NSGD-M, where $n = 1$ and $\delta^{\text{inf}} = 0$, our convergence bound in (5) with $\gamma_0 = \mathcal{O}(1/L_1)$ achieves the $\mathcal{O}\left(\frac{L_1(f(x^0)-f^{\text{inf}})+\sigma+L_0/L_1}{(K+1)^{1/4}}\right)$ convergence, which matches the rate obtained by Hübler et al. (2024, Corollary 3). Unlike the earlier results for single-node NSGD-M, our results extend to multi-node NSGD-M. The bound in (5) for multi-node NSGD-M includes the $\sigma/\sqrt{n}$-term indicating a $\sqrt{n}$-fold reduction in the influence of stochastic variance noise $\sigma$, and the $\gamma_0 L_1^2 \delta^{\text{inf}}$-term accounting for the effect of data heterogeneity.

## 6 EXPERIMENTS

We evaluate the performance of normalized EF21, and compare it against EF21 (Richtárik et al., 2021). We test these algorithms for three nonconvex problems that satisfy generalized smoothness: the problem of minimizing polynomial functions, the logistic regression problem with a nonconvex regularization term over synthetic and benchmark datasets from LIBSVM (Chang & Lin, 2011), and the training of the ResNet-20 (He et al., 2016) model over the CIFAR10 (Krizhevsky, 2009) dataset[1]. For all experiments, we use a top-$k$ sparsifier, which is a $\frac{k}{d}$-contractive compressor.

---

[1] We implemented EF21 and normalized EF21 on training the ResNet-20 model by using PyTorch. Our source codes can be found in the link to error-feedback-generalized-smoothness-paper.

## 6.1 LOGISTIC REGRESSION WITH A NONCONVEX REGULARIZER

First, we consider a logistic regression problem with a nonconvex regularizer, i.e., Problem (1) with

$$f_i(x) = \log(1 + \exp(-b_i a_i^T x)) + \lambda \sum_{j=1}^{d} \frac{x_j^2}{1 + x_j^2},$$

where $a_i \in \mathbb{R}^d$ is the $i^{\text{th}}$ feature vector of data matrix $A \in \mathbb{R}^{n \times d}$ with its class label $b_i \in \{-1, 1\}$, and $\lambda > 0$ is a regularization parameter. Here, $f(x)$ is nonconvex, and $L$-smooth with $L = \|A\|^2 / (4n) + 2\lambda$. Also, each $f_i(x)$ is $L_i$-smooth with $L_i = \|a_i\|^2 / 4 + 2\lambda$, and generalized smooth with $L_0 = 2\lambda + \lambda \sqrt{d} \max_i \|a_i\|$ and $L_1 = \max_i \|a_i\|$. The derivations of smoothness parameters can be found in Appendix F.

In these experiments, we initialized $x^0 \in \mathbb{R}^d$, where each coordinate was drawn from a standard normal distribution $\mathcal{N}(0, 1)$, and set $\lambda = 0.1$. Here, $\lambda > \lambda_{\min}(A^\top A)/(2n)$ to ensure that $f(x)$ is nonconvex. We ran normalized EF21 and EF21 on the following datasets: (1) two from LIB-SVM (Chang & Lin, 2011): Breast Cancer ($n = 683$, $d = 10$, and scaled to $[-1, 1]$), and a1a ($n = 1605$, $d = 123$); and (2) a synthetically generated dataset ($n = 20$, $d = 10$), where the data matrix $A \in \mathbb{R}^{n \times d}$ had entries drawn from $\mathcal{N}(0, 1)$, and the class label $b_i$ was set to either $-1$ or $1$ with equal probability. For EF21, we selected the stepsize $\gamma_k = 1/\left(L + \tilde{L}\sqrt{\beta/\theta}\right)$ with $\tilde{L} = \sqrt{\sum_{i=1}^{n} L_i^2 / n}$, $\theta = 1 - \sqrt{1 - \alpha}$, and $\beta = (1 - \alpha)/(1 - \sqrt{1 - \alpha})$, as given by Richtárik et al. (2021, Theorem 1). For normalized EF21, we chose $\gamma_k = \gamma_0/\sqrt{K + 1}$ with $\gamma_0 > 0$ from Theorem 1, by setting $\gamma_0 = 1$, $K = 100$ for the generated data and Breast Cancer, and $K = 400$ for a1a. We choose $\gamma_0 = 1$, because normalized EF21 with $\gamma_0 \in [1, 10]$ converges faster than that with small values of $\gamma_0$ (e.g. 0.1), when we run the algorithm on a single node ($n = 1$) for minimizing polynomial function and solving logistic regression. We determine $K$ as the smallest number of iterations required to achieve the desired accuracy by performing a grid search with a stepsize of 50.

Figure 2 shows that normalized EF21 outperforms the traditional EF21 on all evaluated datasets, achieving faster convergence and higher solution accuracy. This improvement results from the fact that the theoretical stepsize for normalized EF21, as derived in Theorem 1, is larger than the stepsize for the traditional EF21 outlined by Richtárik et al. (2021, Theorem 1).

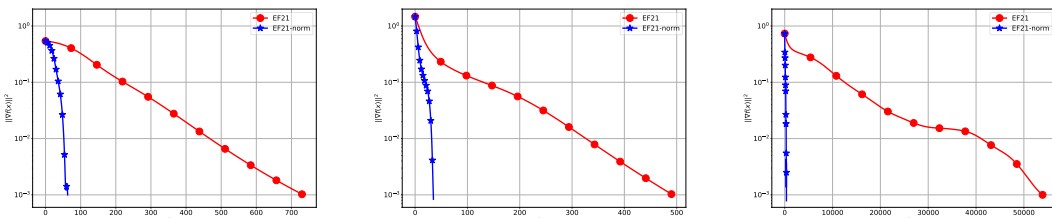

Figure 2: Logistic regression with a nonconvex regularizer using normalized EF21 (EF21-norm) and EF21. We reported $\left\|\nabla f(x^k)\right\|^2$ with respect to iteration count $k$. We used the constant stepsize $\gamma = \frac{1}{L + \tilde{L}\sqrt{\frac{\beta}{\theta}}}$ for EF21, and $\gamma = \frac{\gamma_0}{\sqrt{K+1}}$, $\gamma_0 = 1$ for normalized EF21. Here, $K = 100$ for our generated data (left), and Breast Cancer (middle), while $K = 400$ for a1a (right).

## 6.2 RESNET20 TRAINING OVER CIFAR-10

Next, we trained the ResNet20 (He et al., 2016) model on the CIFAR-10 (Krizhevsky, 2009) dataset, which was demonstrated empirically by Zhang et al. (2020b) to satisfy the $(L_0, L_1)$-smoothness condition. In these experiments, we used a top-$k$ compressor over $50,000$ training images, with evaluation on $10,000$ test images. The dataset was evenly distributed among 5 clients, each using a mini-batch size of 128. Both algorithms were run for 100 epochs with a constant stepsize $\gamma = 5$. Here, one epoch refers to a full pass through the entire dataset processed by all clients.

From Figure 3, under the same constant stepsize and the top-$k$ sparsifier with $k = 0.01d$, normalized EF21 outperforms EF21, in terms of convergence speed (in gradient norms and losses) and accuracy,

relative to the number of bits communicated from each client to the server. Specifically, normalized EF21 achieved accuracy gains of up to 10% over EF21.

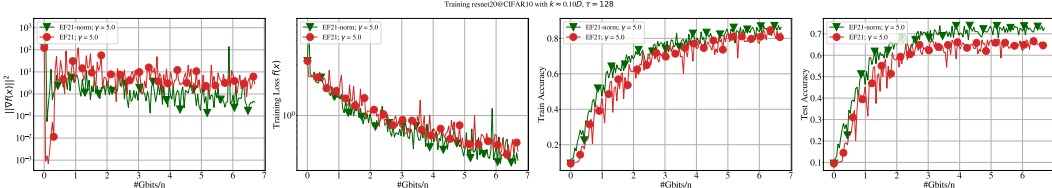

Figure 3: ResNet20 training on CIFAR-10 by using EF21 and normalized EF21 (`EF21-norm`) under the same stepsize $\gamma = 5$ and $k = 0.1d$ for a top-$k$ sparsifier.

## 7 CONCLUSION AND FUTURE WORKS

In this paper, we have demonstrated that normalization can be effectively combined with EF21 to develop distributed error feedback algorithms for solving nonconvex optimization problems under generalized smoothness conditions. Specifically, normalized EF21 and normalized EF21-SGDM achieve convergence rates of $\mathcal{O}(1/K^{1/2})$ in deterministic settings and $\mathcal{O}(1/K^{1/4})$ in stochastic settings, respectively. These convergence rates match those of the vanilla EF21 and EF21-SGDM algorithms. Unlike previous works on distributed algorithms under generalized smoothness, our analysis does not assume data heterogeneity or impose smoothness-dependent restrictions on the stepsize. Finally, our experiments confirm that normalized EF21 exhibits stronger convergence performance compared to the original EF21, due to its larger allowable stepsizes.

Our work implies many promising research directions. One interesting direction is to extend our convergence results for normalized EF21 and normalized EF21-SGDM to accommodate decreasing or adaptive stepsize schedules, as the constant stepsizes required by our current analysis can become impractically small when the total number of iterations is large. In particular, applying appropriate decreasing stepsizes to EF21-SGDM could overcome its current theoretical requirement of a sufficiently large mini-batch size for the stochastic gradient at initialization. Another important direction is the development of distributed and federated algorithms that leverage clipping or normalization for minimizing nonconvex generalized smooth functions.

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

CONTENTS

# A    LEMMAS

In this section, we introduce useful lemmas for our analysis. Lemmas 1 and 2 introduce inequalities by generalized smoothness, while Lemmas 3 and 4 present the descent inequality and convergence rate, respectively, when the normalized gradient descent update is applied.

**Lemma 1.** *Let each $f_i(x)$ be generalized smooth with parameters $L_0, L_1 > 0$, and lower bounded by $f_i^{\mathrm{inf}}$, and let $f(x) = \frac{1}{n} \sum_{i=1}^n f_i(x)$. Then, for any $x, y \in \mathbb{R}^d$*

$$\|\nabla f_i(x) - \nabla f_i(y)\| \leq (L_0 + L_1 \|\nabla f_i(y)\|) \exp(L_1 \|x - y\|) \|x - y\|, \tag{6}$$

$$f_i(y) \leq f_i(x) + \langle \nabla f_i(x), y - x \rangle + \frac{L_0 + L_1 \|\nabla f_i(x)\|}{2} \exp(L_1 \|x - y\|) \|y - x\|^2, \tag{7}$$

$$\frac{\|\nabla f_i(x)\|^2}{4(L_0 + L_1 \|\nabla f_i(x)\|)} \leq f_i(x) - f_i^{\mathrm{inf}}, \text{ and} \tag{8}$$

$$f(y) \leq f(x) + \langle \nabla f(x), y - x \rangle + \frac{L_0 + \frac{L_1}{n} \sum_{i=1}^n \|\nabla f_i(x)\|}{2} \exp(L_1 \|x - y\|) \|y - x\|^2 \tag{9}$$

*Proof.* The first and second statements derive from Chen et al. (2023, Proposition 3.2).

Next, by using the first and second statements, we can derive the third statement. Let us assume that there exists a lower bound of $f_i(x)$, $f_i^{\mathrm{inf}}$, and apply (7) with $y = x - \frac{\nu}{L_0 + L_1 \|\nabla f_i(x)\|} \nabla f_i(x)$ for a given $x \in \mathbb{R}^d$ and $\nu > 0$:

$$
\begin{aligned}
f_i^{\mathrm{inf}} \leq f_i(y) \quad &\leq \quad f_i(x) - \frac{\nu}{L_0 + L_1 \|\nabla f_i(x)\|} \|\nabla f_i(x)\|^2 \\
&\quad + \frac{\nu^2 \|\nabla f_i(x)\|^2}{2(L_0 + L_1 \|\nabla f_i(x)\|)} \exp\left(\frac{L_1 \nu \|\nabla f_i(x)\|}{L_0 + L_1 \|\nabla f_i(x)\|}\right) \\
&\overset{L_0 \geq 0}{\leq} \quad f_i(x) - \frac{\nu}{L_0 + L_1 \|\nabla f_i(x)\|} \|\nabla f_i(x)\|^2 + \frac{\nu \|\nabla f_i(x)\|^2}{2(L_0 + L_1 \|\nabla f_i(x)\|)} \nu \exp(\nu).
\end{aligned}
$$

If $\nu = 1/2$, then $\nu \exp(\nu) \leq 1$, and thus

$$f_i^{\mathrm{inf}} \leq f_i(x) - \frac{1}{4(L_0 + L_1 \|\nabla f_i(x)\|)} \|\nabla f_i(x)\|^2.$$

Finally, we prove the last statement. By the fact that each $f_i(x)$ is symmetric smooth, and $f(x) = \frac{1}{n} \sum_{i=1}^n f_i(x)$,

$$
\begin{aligned}
\|\nabla f(x) - \nabla f(y)\| \quad &\leq \quad \frac{1}{n} \sum_{i=1}^n \|\nabla f_i(x) - \nabla f_i(y)\| \\
&\overset{(6)}{\leq} \quad \frac{1}{n} \sum_{i=1}^n (L_0 + L_1 \|\nabla f_i(y)\|) \exp(L_1 \|x - y\|) \|x - y\| \\
&= \quad \left(L_0 + \frac{L_1}{n} \sum_{i=1}^n \|\nabla f_i(y)\|\right) \exp(L_1 \|x - y\|) \|x - y\|.
\end{aligned}
$$

Hence, for any $y, x \in \mathbb{R}^d$, we have:

$$
\begin{aligned}
&f(y) - f(x) - \langle \nabla f(x), y - x \rangle \\
&= \int_0^1 (\nabla f(y_\theta) - \nabla f(x))^T (y - x) d\theta \\
&\leq \int_0^1 \|\nabla f(y_\theta) - \nabla f(x)\| \|y - x\| d\theta \\
&\leq \int_0^1 \left(L_0 + \frac{L_1}{n} \sum_{i=1}^n \|\nabla f_i(x)\|\right) \exp(L_1 \|y_\theta - x\|) \|y_\theta - x\| \|y - x\| d\theta,
\end{aligned}
$$

where $y_\theta = \theta y + (1 - \theta)x$. From the definition of $y_\theta$, and by the fact that $\exp(\theta y) \leq \exp(y)$ for $\theta \in [0, 1]$,

$$f(y) - f(x) - \langle \nabla f(x), y - x \rangle \leq \int_0^1 \theta \left( L_0 + \frac{L_1}{n} \sum_{i=1}^n \|\nabla f_i(x)\| \right) \exp(L_1 \|y - x\|) \|y - x\|^2 \, d\theta$$

$$= \frac{(L_0 + \frac{L_1}{n} \sum_{i=1}^n \|\nabla f_i(x)\|)}{2} \exp(L_1 \|y - x\|) \|y - x\|^2.$$

$\square$

**Lemma 2.** *Let $f_i(x)$ be generalized smooth with parameters $L_0, L_1 > 0$, and lower bounded by $f_i^{\text{inf}}$, and let $f(x)$ be lower bounded by $f^{\text{inf}}$. Then, for any $x \in \mathbb{R}^d$*

$$\frac{1}{n} \sum_{i=1}^n \|\nabla f_i(x)\| \leq 8L_1(f(x) - f^{\text{inf}}) + \frac{8L_1}{n} \sum_{i=1}^n (f^{\text{inf}} - f_i^{\text{inf}}) + L_0/L_1.$$

*Proof.* By the $(L_0, L_1)$-smoothness of $f_i(x)$,

$$4(f_i(x) - f_i^{\text{inf}}) \overset{(8)}{\geq} \frac{\|\nabla f_i(x)\|^2}{L_0 + L_1 \|\nabla f_i(x)\|} \geq \begin{cases} \frac{\|\nabla f_i(x)\|^2}{2L_0} & \text{if } \|\nabla f_i(x)\| \leq \frac{L_0}{L_1} \\ \frac{\|\nabla f_i(x)\|}{2L_1} & \text{otherwise.} \end{cases}$$

This condition can be equivalently expressed as

$$\begin{aligned} \|\nabla f_i(x)\| &\leq \max(8L_1(f_i(x) - f_i^{\text{inf}}), L_0/L_1) \\ &\leq 8L_1(f_i(x) - f_i^{\text{inf}}) + L_0/L_1 \\ &\leq 8L_1(f_i(x) - f^{\text{inf}}) + 8L_1(f^{\text{inf}} - f_i^{\text{inf}}) + L_0/L_1. \end{aligned}$$

Finally, by the fact that $f(x) = \frac{1}{n} \sum_{i=1}^n f_i(x)$,

$$\frac{1}{n} \sum_{i=1}^n \|\nabla f_i(x)\| \leq 8L_1(f(x) - f^{\text{inf}}) + \frac{8L_1}{n} \sum_{i=1}^n (f^{\text{inf}} - f_i^{\text{inf}}) + L_0/L_1.$$

$\square$

**Lemma 3.** *Consider the problem of minimizing $f(x) = \frac{1}{n} \sum_{i=1}^n f_i(x)$, where each $f_i(x)$ is generalized smooth with parameters $L_0, L_1 > 0$. Let $x^{k+1} = x^k - \frac{\gamma_k}{\|v^k\|} v^k$ for $\gamma_k > 0$. Then,*

$$\begin{aligned} f(x^{k+1}) &\leq f(x^k) - \gamma_k \|\nabla f(x^k)\| + 2\gamma_k \|\nabla f(x^k) - v^k\| + \\ &\quad + \frac{\gamma_k^2}{2} \exp(\gamma_k L_1) \left( L_0 + \frac{L_1}{n} \sum_{i=1}^n \|\nabla f_i(x^k)\| \right). \end{aligned}$$

*Proof.* Let each $f_i(x)$ be generalized smooth with $L_0, L_1 > 0$, and $f(x) = \frac{1}{n} \sum_{i=1}^n f_i(x)$. By (9) of Lemma 1, and by the fact that $x^{k+1} = x^k - \frac{\gamma_k}{\|v^k\|} v^k$ for $\gamma_k > 0$,

$$\begin{aligned} f(x^{k+1}) &\leq f(x^k) - \frac{\gamma_k}{\|v^k\|} \langle \nabla f(x^k), v^k \rangle + \frac{\gamma_k^2}{2} \exp(\gamma_k L_1) \left( L_0 + \frac{L_1}{n} \sum_{i=1}^n \|\nabla f_i(x^k)\| \right) \\ &= f(x^k) - \frac{\gamma_k}{\|v^k\|} \langle \nabla f(x^k) - v^k, v^k \rangle - \gamma_k \|v^k\| \\ &\quad + \frac{\gamma_k^2}{2} \exp(\gamma_k L_1) \left( L_0 + \frac{L_1}{n} \sum_{i=1}^n \|\nabla f_i(x^k)\| \right) \\ &\leq f(x^k) + \gamma_k \|\nabla f(x^k) - v^k\| - \gamma_k \|v^k\| \\ &\quad + \frac{\gamma_k^2}{2} \exp(\gamma_k L_1) \left( L_0 + \frac{L_1}{n} \sum_{i=1}^n \|\nabla f_i(x^k)\| \right), \end{aligned}$$

where we reach the last inequality by Cauchy-Schwartz inequality. Next, since

$$- \left\| v^k \right\| \overset{\text{triangle ineq.}}{\leq} - \left\| \nabla f(x^k) \right\| + \left\| \nabla f(x^k) - v^k \right\|,$$

we get

$$
\begin{aligned}
f(x^{k+1}) &\leq f(x^k) - \gamma_k \left\| \nabla f(x^k) \right\| + 2\gamma_k \left\| \nabla f(x^k) - v^k \right\| \\
&\quad + \frac{\gamma_k^2}{2} \exp(\gamma_k L_1) \left( L_0 + \frac{L_1}{n} \sum_{i=1}^n \left\| \nabla f_i(x^k) \right\| \right).
\end{aligned}
$$

$\square$

**Lemma 4.** *Let $V^k, W^k$ be non-negative sequences satisfying*

$$V^{k+1} \leq (1 + b_1 \exp(L_1\gamma)\gamma^2)V^k - b_2\gamma W^k + b_3 \exp(L_1\gamma)\gamma^2,$$

*for $\gamma, b_1, b_2, b_3 > 0$. Then,*

$$\min_{k=0,1,\ldots,K} W^k \leq \frac{V^0 \exp(b_1 \exp(L_1\gamma)\gamma^2 (K+1))}{b_2\gamma(K+1)} + \frac{b_3}{b_2} \exp(L_1\gamma)\gamma.$$

*Proof.* Define $\beta_k = \frac{\beta_{k-1}}{1+b_1 \exp(L_1\gamma)\gamma^2}$ for $k = 0, 1, \ldots$ and $\beta_{-1} = 1$. Then, we can show that $\beta_k = \frac{1}{(1+b_1 \exp(L_1\gamma)\gamma^2)^{k+1}}$ for $k = 0, 1, \ldots$, and that

$$
\begin{aligned}
\beta_k V^{k+1} &\leq (1 + b_1 \exp(L_1\gamma)\gamma^2)\beta_k V^k - b_2\gamma\beta_k W^k + b_3 \exp(L_1\gamma)\gamma^2\beta_k \\
&= \beta_{k-1} V^k - b_2\gamma\beta_k W^k + b_3 \exp(L_1\gamma)\gamma^2\beta_k.
\end{aligned}
$$

Therefore,

$$
\begin{aligned}
\min_{k=0,1,\ldots,K} W^k &\leq \frac{1}{\sum_{k=0}^K \beta_k} \sum_{k=0}^K \beta_k W^k \\
&\leq \frac{\sum_{k=0}^K (\beta_{k-1} V^k - \beta_k V^{k+1})}{b_2\gamma \sum_{k=0}^K \beta_k} + \frac{b_3}{b_2} \exp(L_1\gamma)\gamma \\
&= \frac{\beta_{-1} V^0 - \beta_K V^{k+1}}{b_2\gamma \sum_{k=0}^K \beta_k} + \frac{b_3}{b_2} \exp(L_1\gamma)\gamma.
\end{aligned}
$$

By the fact that $\beta_{-1} = 1$, $\beta_K > 0$, and $V^{k+1} \geq 0$,

$$\min_{k=0,1,\ldots,K} W^k \leq \frac{V^0}{b_2\gamma \sum_{k=0}^K \beta_k} + \frac{b_3}{b_2} \exp(L_1\gamma)\gamma.$$

Next, since

$$\sum_{k=0}^K \beta_k \geq (K+1) \min_{k=0,1,\ldots,K} \beta_k = \frac{K+1}{(1+b_1 \exp(L_1\gamma)\gamma^2)^{K+1}},$$

we have

$$
\begin{aligned}
\min_{k=0,1,\ldots,K} W^k &\leq \frac{V^0(1 + b_1 \exp(L_1\gamma)\gamma^2)^{K+1}}{b_2\gamma(K+1)} + \frac{b_3}{b_2} \exp(L_1\gamma)\gamma \\
&\overset{1+x\leq\exp(x)}{\leq} \frac{V^0 \exp(b_1 \exp(L_1\gamma)\gamma^2 (K+1))}{b_2\gamma(K+1)} + \frac{b_3}{b_2} \exp(L_1\gamma)\gamma.
\end{aligned}
$$

$\square$

## B  CONVERGENCE PROOF OF NORMALIZED EF21 (THEOREM 1)

In this section, we derive the convergence rate results of normalized EF21. To prove this, we present the following descent lemma for normalized EF21.

**Lemma 5.** *Consider Problem (1), where Assumption 1 (lower bound of $f$), Assumption 2 (lower bound of $f_i$), Assumption 3 (generalized smooth of $f_i$), and Assumption 4 ($\alpha$-contractive property of $\mathcal{C}^k$) hold. Then, the iterates $\{x^k\}$ generated by normalized EF21 (Algorithm 1) satisfy*

$$\mathrm{E}\left[V^{k+1}\right] \le \mathrm{E}\left[V^k\right] + c_1\gamma_k^2\frac{1}{n}\sum_{i=1}^n \mathrm{E}\left[\left\|\nabla f_i(x^k)\right\|\right] - \gamma_k\mathrm{E}\left[\left\|\nabla f(x^k)\right\|\right] + c_0\gamma_k^2,$$

*where $V^k := f(x^k) - f^{\inf} + \frac{2\gamma_k}{1-\sqrt{1-\alpha}}\frac{1}{n}\sum_{i=1}^n \left\|\nabla f_i(x^k) - v_i^k\right\|$, and $c_i = \frac{L_i}{2} + 2\frac{\sqrt{1-\alpha}L_i}{1-\sqrt{1-\alpha}}$ for $i = 0, 1$.*

*Proof.* We prove the result in two steps.

**Step 1) Bound** $\mathrm{E}\left[\left\|\nabla f_i(x^{k+1}) - v_i^{k+1}\right\|\right]$**.**  From the definition of the Euclidean norm, and by taking the expectation conditioned on $x^{k+1}, v_i^k$, and by the update of $v_i^k$ from Algorithm 1

$$\mathrm{E}\left[\left\|\nabla f_i(x^{k+1}) - v_i^{k+1}\right\| \big| x^{k+1}, v_i^k\right]$$
$$= \mathrm{E}\left[\left\|\nabla f_i(x^{k+1}) - v_i^k - \mathcal{C}^k(\nabla f_i(x^{k+1}) - v_i^k)\right\| \big| x^{k+1}, v_i^k\right]$$
$$\le \sqrt{\mathrm{E}\left[\left\|\nabla f_i(x^{k+1}) - v_i^k - \mathcal{C}(\nabla f_i(x^{k+1}) - v_i^k)\right\|^2 \big| x^{k+1}, v_i^k\right]},$$

where we use the concave property of the square root function, and Jensen's inequality for the concave function, i.e., $\mathrm{E}\left[f(x)\right] \le f(\mathrm{E}\left[x\right])$ if $f(x)$ is concave.

By the $\alpha$-contractive property of compressors in (3), by the fact that $\left\|\nabla f_i(x^{k+1}) - v_i^k\right\|$ is a constant conditioned on $x^{k+1}, v_i^k$, and then by the triangle inequality,

$$\mathrm{E}\left[\left\|\nabla f_i(x^{k+1}) - v_i^{k+1}\right\| \big| x^{k+1}, v_i^k\right] \le \sqrt{(1-\alpha)\mathrm{E}\left[\left\|\nabla f_i(x^{k+1}) - v_i^k\right\|^2 \big| x^{k+1}, v_i^k\right]}$$
$$= \sqrt{1-\alpha}\left\|\nabla f_i(x^{k+1}) - v_i^k\right\|$$
$$\le \sqrt{1-\alpha}\left\|\nabla f_i(x^k) - v_i^k\right\| + \sqrt{1-\alpha}\left\|\nabla f_i(x^{k+1}) - \nabla f_i(x^k)\right\|.$$

By the generalized smoothness of $f_i(x)$ in (2), and by the fact that $x^{k+1} = x^k - \gamma_k\frac{v^k}{\|v^k\|}$,

$$\mathrm{E}\left[\left\|\nabla f_i(x^{k+1}) - v_i^{k+1}\right\| \big| x^{k+1}, v_i^k\right] \le \sqrt{1-\alpha}\left\|\nabla f_i(x^k) - v_i^k\right\|$$
$$+ \sqrt{1-\alpha}(L_0 + L_1\left\|\nabla f_i(x^k)\right\|)\exp(L_1\gamma_k)\gamma_k.$$

Let $\gamma_k > 0$ be constants conditioned on $x^{k+1}, v_i^k$. Then, by the tower property, i.e.,

$$\mathrm{E}\left[\left\|\nabla f_i(x^{k+1}) - v_i^{k+1}\right\|\right] = \mathrm{E}\left[\mathrm{E}\left[\left\|\nabla f_i(x^{k+1}) - v_i^{k+1}\right\| \big| x^{k+1}, v_i^k\right]\right],$$

we have

$$\mathrm{E}\left[\left\|\nabla f_i(x^{k+1}) - v_i^{k+1}\right\|\right] \le \sqrt{1-\alpha}\mathrm{E}\left[\left\|\nabla f_i(x^k) - v_i^k\right\|\right]$$
$$+ \sqrt{1-\alpha}\exp(L_1\gamma_k)\gamma_k(L_0 + L_1\mathrm{E}\left[\left\|\nabla f_i(x^k)\right\|\right]). \quad (10)$$

**Step 2) Bound** $V^k := f(x^k) - f^{\inf} + A_k\frac{1}{n}\sum_{i=1}^n \left\|\nabla f_i(x^k) - v_i^k\right\|$ **for some** $A_k > 0$**.**  Next, define $V^k := f(x^k) - f^{\inf} + A_k\frac{1}{n}\sum_{i=1}^n \left\|\nabla f_i(x^k) - v_i^k\right\|$ for some positive constants $A_k$. Then, from

Lemma 3,

$$
\begin{aligned}
\mathrm{E}\left[V^{k+1}\right] \quad \leq \quad & \mathrm{E}\left[f(x^k) - f^{\inf}\right] - \gamma_k \mathrm{E}\left[\left\|\nabla f(x^k)\right\|\right] \\
& + \exp(L_1\gamma_k)\gamma_k^2 \frac{L_1}{2n} \sum_{i=1}^n \mathrm{E}\left[\left\|\nabla f_i(x^k)\right\|\right] + \exp(L_1\gamma_k)\gamma_k^2 \frac{L_0}{2} \\
& + 2\gamma_k \mathrm{E}\left[\left\|\nabla f(x^k) - v^k\right\|\right] + A_{k+1}\frac{1}{n}\sum_{i=1}^n \mathrm{E}\left[\left\|\nabla f_i(x^{k+1}) - v_i^{k+1}\right\|\right].
\end{aligned}
$$

By the fact that $\nabla f(x^k) = \frac{1}{n}\sum_{i=1}^n \nabla f_i(x^k)$, and $v^k = \frac{1}{n}\sum_{i=1}^n v_i^k$, and by the triangle inequality,

$$
\begin{aligned}
\mathrm{E}\left[V^{k+1}\right] \quad \leq \quad & \mathrm{E}\left[f(x^k) - f^{\inf}\right] - \gamma_k \mathrm{E}\left[\left\|\nabla f(x^k)\right\|\right] \\
& + \exp(L_1\gamma_k)\gamma_k^2 \frac{L_1}{2n} \sum_{i=1}^n \mathrm{E}\left[\left\|\nabla f_i(x^k)\right\|\right] + \exp(L_1\gamma_k)\gamma_k^2 \frac{L_0}{2} \\
& + 2\gamma_k \frac{1}{n}\sum_{i=1}^n \mathrm{E}\left[\left\|\nabla f_i(x^k) - v_i^k\right\|\right] + A_{k+1}\frac{1}{n}\sum_{i=1}^n \mathrm{E}\left[\left\|\nabla f_i(x^{k+1}) - v_i^{k+1}\right\|\right].
\end{aligned}
$$

Next, by (10),

$$
\begin{aligned}
\mathrm{E}\left[V^{k+1}\right] \quad \leq \quad & \mathrm{E}\left[f(x^k) - f^{\inf}\right] - \gamma_k \mathrm{E}\left[\left\|\nabla f(x^k)\right\|\right] + \left(\frac{\gamma_k^2}{2} + A_{k+1}\sqrt{1-\alpha}\gamma_k\right)\exp(L_1\gamma_k)L_0 \\
& + \left(\frac{\gamma_k^2}{2} + A_{k+1}\sqrt{1-\alpha}\gamma_k\right)\exp(L_1\gamma_k)L_1 \frac{1}{n}\sum_{i=1}^n \mathrm{E}\left[\left\|\nabla f_i(x^k)\right\|\right] \\
& + \left(2\gamma_k + A_{k+1}\sqrt{1-\alpha}\right)\frac{1}{n}\sum_{i=1}^n \mathrm{E}\left[\left\|\nabla f_i(x^k) - v_i^k\right\|\right].
\end{aligned}
$$

If $A_k = \frac{2\gamma_k}{1-\sqrt{1-\alpha}}$, and $\gamma_k$ satisfies $\gamma_{k+1} \leq \gamma_k$, then

$$
2\gamma_k + A_{k+1}\sqrt{1-\alpha} \leq 2\gamma_k + A_k\sqrt{1-\alpha} = A_k.
$$

Therefore,

$$
\begin{aligned}
\mathrm{E}\left[V^{k+1}\right] \quad \leq \quad & \mathrm{E}\left[V^k\right] + c_1\exp(L_1\gamma_k)\gamma_k^2 \frac{1}{n}\sum_{i=1}^n \mathrm{E}\left[\left\|\nabla f_i(x^k)\right\|\right] \\
& - \gamma_k \mathrm{E}\left[\left\|\nabla f(x^k)\right\|\right] + c_0\exp(L_1\gamma_k)\gamma_k^2,
\end{aligned}
$$

where $c_i = \frac{L_i}{2} + 2\frac{\sqrt{1-\alpha}L_i}{1-\sqrt{1-\alpha}}$ for $i = 0, 1$. $\qquad\square$

## B.1 CONVERGENCE PROOF FOR THEOREM 1

Now, we are ready to prove Theorem 1. From Lemma 5 and 2, and by the fact that $c_1 L_0/L_1 = c_0$

$$
\mathrm{E}\left[V^{k+1}\right] \leq \mathrm{E}\left[V^k\right] + 8c_1 L_1 \exp(L_1\gamma_k)\gamma_k^2 \mathrm{E}\left[f(x^k) - f^{\inf}\right] - \gamma_k \mathrm{E}\left[\left\|\nabla f(x^k)\right\|\right] + B\exp(L_1\gamma_k)\gamma_k^2,
$$

where $B = 2c_0 + \frac{8c_1 L_1}{n}\sum_{i=1}^n (f^{\inf} - f_i^{\inf})$. By the fact that $f(x^k) - f^{\inf} \leq V^k$,

$$
\mathrm{E}\left[V^{k+1}\right] \leq (1 + 8c_1 L_1 \exp(L_1\gamma_k)\gamma_k^2)\mathrm{E}\left[V^k\right] - \gamma_k \mathrm{E}\left[\left\|\nabla f(x^k)\right\|\right] + B\exp(L_1\gamma_k)\gamma_k^2.
$$

By applying Lemma 4 with $V^k = \mathrm{E}\left[V^k\right]$, $W^k = \mathrm{E}\left[\left\|\nabla f(x^k)\right\|\right]$, $b_1 = 8c_1 L_1$, $b_2 = 1$, and $b_3 = B$,

$$
\min_{k=0,1,\ldots,K} W^k \leq \frac{V^0 \exp(b_1 \exp(L_1\gamma)\gamma^2(K+1))}{b_2\gamma(K+1)} + \frac{b_3}{b_2}\exp(L_1\gamma)\gamma.
$$

Finally, if $\gamma = \frac{\gamma_0}{\sqrt{K+1}}$ with $\gamma_0 > 0$, then $\exp(L_1\gamma_k) \leq \exp(L_1\gamma_0)$, and thus

$$
\min_{k=0,1,\ldots,K} W^k \leq \frac{V^0 \exp(b_1 \exp(L_1\gamma_0)\gamma_0^2)}{b_2\gamma_0\sqrt{K+1}} + \frac{b_3}{b_2}\frac{\gamma_0 \exp(L_1\gamma_0)}{\sqrt{K+1}}.
$$

## C  CONVERGENCE OF NORMALIZED EF21 FOR A SINGLE-NODE CASE

In this section, we provide the convergence of normalized EF21 for a single-node case. In particular, the algorithm enjoys the $\mathcal{O}(1/K)$ convergence up to the additive constant $\frac{c_0\gamma}{1-c_1\exp(L_1\gamma)\gamma}$. In contrast to Theorem 1 for multi-node normalized EF21, the next result for single-node normalized EF21 applies for any $\gamma_k = \gamma \in (0, 1/(\beta c_1))$ with $\beta \geq 2$, $c_1 = \frac{L_1}{2} + 2\frac{\sqrt{1-\alpha}L_1}{1-\sqrt{1-\alpha}}$, and $\alpha \in (0, 1]$.

**Theorem 3.** *Consider the problem of minimizing $f(x)$, which satisfies Assumption 1 (lower bound of $f$), and Assumption 3 (generalized smoothness of $f$). Further, let Assumption 4 (contractive compressor) hold. Then, the iterates $\{x^k\}$ generated by normalized EF21 (Algorithm 1) with $n = 1$, $\gamma_k = \gamma = 1/(\beta c_1)$ and $\beta \geq 2$ satisfy*

$$\min_{k=0,1,\ldots,K} \mathrm{E}\left[\left\|\nabla f(x^k)\right\|\right] \leq \frac{\mathrm{E}\left[V^0\right] - \mathrm{E}\left[V^{K+1}\right]}{\gamma(1 - c_1\exp(L_1\gamma)\gamma)(K+1)} + \frac{c_0\gamma}{1 - c_1\exp(L_1\gamma)\gamma},$$

*where $V^k = f(x^k) - f^{\inf} + \frac{2\gamma}{1-\sqrt{1-\alpha}}\left\|\nabla f(x^k) - v^k\right\|$, and $c_i = \frac{L_i}{2} + 2\frac{\sqrt{1-\alpha}L_i}{1-\sqrt{1-\alpha}}$ for $i = 0, 1$.*

*Proof.* We prove the result in the following steps:

**Step 1) Bound** $\mathrm{E}\left[\left\|\nabla f(x^{k+1}) - v^{k+1}\right\|\right]$**.** From the definition of the Euclidean norm, and by taking the expectation conditioned on $x^{k+1}, v^k$,

$$\mathrm{E}\left[\left\|\nabla f(x^{k+1}) - v^{k+1}\right\|\big|\, x^{k+1}, v^k\right] \stackrel{v^k}{=} \mathrm{E}\left[\left\|\nabla f(x^{k+1}) - v^k - \mathcal{C}(\nabla f(x^{k+1}) - v^k)\right\|\big|\, x^{k+1}, v^k\right]$$

$$\leq \sqrt{\mathrm{E}\left[\left\|\nabla f(x^{k+1}) - v^k - \mathcal{C}(\nabla f(x^{k+1}) - v^k)\right\|^2\Big|\, x^{k+1}, v^k\right]}$$

$$\stackrel{(3)}{\leq} \sqrt{(1-\alpha)\mathrm{E}\left[\left\|\nabla f(x^{k+1}) - v^k\right\|^2\Big|\, x^{k+1}, v^k\right]}$$

$$= \sqrt{1-\alpha}\left\|\nabla f(x^{k+1}) - v^k\right\|,$$

where we reach the second inequality by the fact that the square root function is concave, and the last inequality by the fact that $\left\|\nabla f(x^{k+1}) - v^k\right\|$ is a constant conditioned on $x^{k+1}, v^k$. Next, by the triangle inequality,

$$\mathrm{E}\left[\left\|\nabla f(x^{k+1}) - v^{k+1}\right\|\big|\, x^{k+1}, v^k\right] \leq \sqrt{1-\alpha}\left\|\nabla f(x^k) - v^k\right\| + \sqrt{1-\alpha}\left\|\nabla f(x^{k+1}) - \nabla f(x^k)\right\|$$

$$\stackrel{(6)}{\leq} \sqrt{1-\alpha}\left\|\nabla f(x^k) - v^k\right\| + \sqrt{1-\alpha}(L_0 + L_1\left\|\nabla f(x^k)\right\|)\exp\left(L_1\left\|x^{k+1} - x^k\right\|\right)\left\|x^{k+1} - x^k\right\|$$

$$\stackrel{x^{k+1}}{\leq} \sqrt{1-\alpha}\left\|\nabla f(x^k) - v^k\right\| + \sqrt{1-\alpha}(L_0 + L_1\left\|\nabla f(x^k)\right\|)\exp(L_1\gamma_k)\gamma_k.$$

Next, by the tower property, and by the fact that $\{\gamma_k\}$ are constants,

$$\mathrm{E}\left[\left\|\nabla f(x^{k+1}) - v^{k+1}\right\|\right] = \mathrm{E}\left[\mathrm{E}\left[\left\|\nabla f(x^{k+1}) - v^{k+1}\right\|\big|\, x^{k+1}, v^k\right]\right]$$

$$\leq \sqrt{1-\alpha}\mathrm{E}\left[\left\|\nabla f(x^k) - v^k\right\|\right] + \sqrt{1-\alpha}(L_0 + L_1\mathrm{E}\left[\left\|\nabla f(x^k)\right\|\right])\exp(L_1\gamma_k)\gamma_k. \quad (11)$$

**Step 2) Bound** $V^k := f(x^k) - f^{\inf} + A_k\left\|\nabla f(x^k) - v^k\right\|$ **for some** $A_k > 0$**.** Denote $V^k := f(x^k) - f^{\inf} + A_k\left\|\nabla f(x^k) - v^k\right\|$ for some constants $A_k > 0$. Then, from the definition of $V^{k+1}$, from Lemma 3 with $n = 1$, and by the fact $f(x)$ is generalized smooth,

$$\mathrm{E}\left[V^{k+1}\right] \leq \mathrm{E}\left[f(x^k) - f^{\inf}\right] - \left(\gamma_k - \frac{\gamma_k^2 L_1}{2}\exp(L_1\gamma_k)\right)\mathrm{E}\left[\left\|\nabla f(x^k)\right\|\right] + \frac{\gamma_k^2 L_0}{2}\exp(L_1\gamma_k)$$

$$+ 2\gamma_k\mathrm{E}\left[\left\|\nabla f(x^k) - v^k\right\|\right] + A_{k+1}\mathrm{E}\left[\left\|\nabla f(x^{k+1}) - v^{k+1}\right\|\right]$$

$$\stackrel{(11)}{\leq} \mathrm{E}\left[f(x^k) - f^{\inf}\right] + \left(2\gamma_k + A_{k+1}\sqrt{1-\alpha}\right)\mathrm{E}\left[\left\|\nabla f(x^k) - v^k\right\|\right]$$

$$- \left(\gamma_k - \frac{\gamma_k^2 L_1}{2}\exp(L_1\gamma_k) - A_{k+1}\sqrt{1-\alpha}L_1\gamma_k\exp(L_1\gamma_k)\right)\mathrm{E}\left[\left\|\nabla f(x^k)\right\|\right]$$

$$+ \frac{\gamma_k^2 L_0}{2}\exp(L_1\gamma_k) + A_{k+1}\sqrt{1-\alpha}L_0\gamma_k\exp(L_1\gamma_k)$$

If $A_k = \frac{2\gamma_k}{1-\sqrt{1-\alpha}}$ and $\gamma_k$ satisfies $\gamma_{k+1} \le \gamma_k$, then

$$2\gamma_k + A_{k+1}\sqrt{1-\alpha} \le 2\gamma_k + A_k\sqrt{1-\alpha} = A_k.$$

Therefore,

$$\mathrm{E}\left[V^{k+1}\right] \le \mathrm{E}\left[V^k\right] - \left(\gamma_k - c_1 \exp(L_1\gamma_k)\gamma_k^2\right)\mathrm{E}\left[\left\|\nabla f(x^k)\right\|\right] + c_0 \exp(L_1\gamma_k)\gamma_k^2,$$

where $c_i = \frac{L_i}{2} + 2\frac{\sqrt{1-\alpha}L_i}{1-\sqrt{1-\alpha}}$ for $i = 0, 1$.

**Step 3) Complete the convergence bound.** If $\gamma_k = \gamma = 1/(\beta c_1)$ for $\beta \ge 2$, then $c_1 \exp(L_1\gamma)\gamma = \exp(L_1/(\beta c_1))/\beta \le \exp(2/\beta)/\beta \le 0.7 < 1$, and

$$\mathrm{E}\left[V^{k+1}\right] \quad \le \quad \mathrm{E}\left[V^k\right] - \gamma\left(1 - c_1\exp(L_1\gamma)\gamma\right)\mathrm{E}\left[\left\|\nabla f(x^k)\right\|\right] + c_0\gamma^2.$$

By re-arranging the terms,

$$
\begin{aligned}
\min_{k=0,1,\dots,K}\mathrm{E}\left[\left\|\nabla f(x^k)\right\|\right] \quad &\le \quad \frac{1}{K+1}\sum_{k=0}^{K}\mathrm{E}\left[\left\|\nabla f(x^k)\right\|\right] \\
&\le \quad \frac{\mathrm{E}\left[V^0\right] - \mathrm{E}\left[V^{K+1}\right]}{\gamma(1 - c_1\exp(L_1\gamma)\gamma)(K+1)} + \frac{c_0\gamma}{1 - c_1\exp(L_1\gamma)\gamma}.
\end{aligned}
$$

By the fact $V^k \ge 0$, we complete the proof. $\qquad\square$

## D    Convergence of Normalized EF21-SGDM (Theorem 2)

In this section, we derive the convergence rate results of normalized EF21-SGDM. We first introduce auxiliary lemmas in Section D.1, and later prove the convergence theorem (Theorem 2) in Section D.2.

### D.1    Auxiliary Lemmas

Now, we provide useful lemmas for analyzing EF21-SGDM. First, Lemma 6 shows the descent inequality of the normalized gradient descent update under Assumption 3 (generalized smoothness of $f_i$). Second, Lemma 7 and 8 provide the upper-bound of the Euclidean distance between $v_i^k$ and $g_i^k$, and of the Euclidean distance between $v_i^k$ and $\nabla f_i(x^k)$, respectively. Third, Lemma 9 presents the convergence rate from the recursion of the non-negative sequences $r^k, s^k$.

**Lemma 6.** *Consider the iterates $\{x^k\}$ generated by Algorithm 2. If Assumption 3 holds, then for any $\gamma_k > 0, \eta_k \in [0, 1]$,*

$$
\begin{aligned}
f(x^{k+1}) \;\leq\; & f(x^k) - \gamma_k \left\| \nabla f(x^k) \right\| + 2\gamma_k \left\| \nabla f(x^k) - v^k \right\| + 2\gamma_k \left\| v^k - g^k \right\| \\
& + \frac{\gamma_k^2}{2} \exp\left(\gamma_k L_1\right) \left( L_0 + \frac{L_1}{n} \sum_{i=1}^n \left\| \nabla f_i(x^k) \right\| \right).
\end{aligned}
$$

*Proof.* By applying the triangle inequality into Lemma 3, we complete the proof.    □

**Lemma 7.** *Consider the iterates $\{x^k\}$ generated by Algorithm 2. If Assumptions 3, 4, and 5 hold, then for $\gamma_k > 0, \eta_k \in [0, 1]$, and $k \geq 0$,*

$$
\begin{aligned}
\frac{1}{n} \sum_{i=1}^n \mathrm{E}\left[\left\| v_i^{k+1} - g_i^{k+1} \right\|\right] \leq & \frac{\sqrt{1-\alpha}}{n} \sum_{i=1}^n \mathrm{E}\left[\left\| v_i^k - g_i^k \right\|\right] + \frac{\sqrt{1-\alpha}\eta_{k+1}}{n} \sum_{i=1}^n \mathrm{E}\left[\left\| v_i^k - \nabla f_i(x^k) \right\|\right] \\
& + \sqrt{1-\alpha}\eta_{k+1}\gamma_k \exp\left(\gamma_k L_1\right)\left( L_0 + L_1 \frac{1}{n} \sum_{i=1}^n \mathrm{E}\left[\left\| \nabla f_i(x^k) \right\|\right] \right) \\
& + \sqrt{1-\alpha}\eta_k\sigma.
\end{aligned}
$$

*Proof.* Taking conditional expectation by $\mathcal{F}_{k+1} = \{v_i^{k+1}, x^{k+1}, g_i^k\}$, using the concave property of the squared root of the function, and applying the definition of $g_i^k$ in Algorithm 2, we have

$$
\begin{aligned}
\mathrm{E}\left[\left\| v_i^{k+1} - g_i^{k+1} \right\| \,\middle|\, \mathcal{F}_{k+1}\right] \;\leq\; & \sqrt{\mathrm{E}\left[\left\| v_i^{k+1} - g_i^{k+1} \right\|^2 \,\middle|\, \mathcal{F}_{k+1}\right]} \\
=\; & \sqrt{\mathrm{E}\left[\left\| v_i^{k+1} - g_i^k - \mathcal{C}^k\left(v_i^{k+1} - g_i^k\right) \right\|^2 \,\middle|\, \mathcal{F}_{k+1}\right]} \\
\overset{(3)}{\leq}\; & \sqrt{\mathrm{E}\left[(1-\alpha) \left\| v_i^{k+1} - g_i^k \right\|^2 \,\middle|\, \mathcal{F}_{k+1}\right]}.
\end{aligned}
$$

Next, let $\gamma_k = \gamma > 0$, and $\eta_k = \eta \in [0, 1]$. By the fact that $v_i^{k+1}, g_i^k$ are constants being conditioned on $\mathcal{F}_{k+1}$, and by the triangle inequality,

$$
\begin{aligned}
\mathrm{E}\left[\left\| v_i^{k+1} - g_i^{k+1} \right\| \,\middle|\, \mathcal{F}_{k+1}\right] \;\leq\; & \sqrt{1-\alpha} \left\| v_i^k - g_i^k \right\| + \sqrt{1-\alpha} \left\| v_i^{k+1} - v_i^k \right\| \\
=\; & \sqrt{1-\alpha} \left\| v_i^k - g_i^k \right\| + \sqrt{1-\alpha}\eta_{k+1} \left\| \nabla f(x^{k+1}; \xi_i^{k+1}) - v_i^k \right\|.
\end{aligned}
$$

Here, the equality comes from the definition of $v_i^{k+1}$ in Algorithm 2. Next, by the triangle inequality,

$$
\begin{aligned}
\mathrm{E}\left[\left\|v_i^{k+1} - g_i^{k+1}\right\| \middle| \mathcal{F}_{k+1}\right] &\leq \sqrt{1-\alpha}\left\|v_i^k - g_i^k\right\| + \sqrt{1-\alpha}\eta_{k+1}\|v_i^k - \nabla f_i(x^k)\| \\
&\quad + \sqrt{1-\alpha}\eta_{k+1}\left\|\nabla f_i(x^k) - \nabla f_i(x^{k+1})\right\| \\
&\quad + \sqrt{1-\alpha}\eta_{k+1}\left\|\nabla f_i(x^{k+1};\xi_i^{k+1}) - \nabla f_i(x^{k+1})\right\| \\
&\overset{(6)}{\leq} \sqrt{1-\alpha}\left\|v_i^k - g_i^k\right\| + \sqrt{1-\alpha}\eta_{k+1}\|v_i^k - \nabla f_i(x^k)\| \\
&\quad + \sqrt{1-\alpha}\eta_{k+1}\left(L_0 + L_1\left\|\nabla f_i(x^k)\right\|\right)\exp\left(L_1\left\|x^{k+1}-x^k\right\|\right)\left\|x^{k+1}-x^k\right\| \\
&\quad + \sqrt{1-\alpha}\eta_{k+1}\left\|\nabla f(x^{k+1};\xi_i^{k+1}) - \nabla f(x^{k+1})\right\|.
\end{aligned}
$$

Next, using $x^{k+1} - x^k = -\gamma_k \frac{g^k}{\|g^k\|}$, and taking an expectation, we obtain

$$
\begin{aligned}
\mathrm{E}\left[\left\|v_i^{k+1} - g_i^{k+1}\right\|\right] &\leq \sqrt{1-\alpha}\mathrm{E}\left[\left\|v_i^k - g_i^k\right\|\right] + \sqrt{1-\alpha}\eta_{k+1}\mathrm{E}\left[\left\|v_i^k - \nabla f_i(x^k)\right\|\right] \\
&\quad + \sqrt{1-\alpha}\eta_{k+1}\gamma_k\exp\left(\gamma_k L_1\right)\left(L_0 + L_1\mathrm{E}\left[\left\|\nabla f_i(x^k)\right\|\right]\right) \\
&\quad + \sqrt{1-\alpha}\eta_{k+1}\mathrm{E}\left[\left\|\nabla f_i(x^{k+1};\xi_i^{k+1}) - \nabla f_i(x^{k+1})\right\|\right].
\end{aligned}
$$

Finally, since

$$
\mathrm{E}\left[\left\|\nabla f_i(x^{k+1};\xi_i^{k+1}) - \nabla f_i(x^{k+1})\right\|\right] \leq \sqrt{\mathrm{E}\left[\left\|\nabla f_i(x^{k+1};\xi_i^{k+1}) - \nabla f_i(x^{k+1})\right\|^2\right]}
$$

$$
\overset{(4)}{\leq} \sigma,
$$

we can obtain the upper bound for $\frac{1}{n}\sum_{i=1}^n \mathrm{E}\left[\left\|v_i^{k+1} - g_i^{k+1}\right\|\right]$. $\qquad\square$

**Lemma 8.** *Consider the iterates $\{x^k\}$ generated by Algorithm 2. If Assumptions 3, and 5 hold, then for any $\gamma_k \equiv \gamma > 0$, $\eta_k \equiv \eta$, and $k \geq 0$,*

$$
\begin{aligned}
\mathrm{E}\left[\left\|v^k - \nabla f(x^k)\right\|\right] &\leq (1-\eta)^k\mathrm{E}\left[\left\|v^0 - \nabla f(x^0)\right\|\right] + \frac{\sqrt{\eta}\sigma}{\sqrt{n}} + \frac{\gamma}{\eta}L_0\exp\left(\gamma L_1\right) \\
&\quad + \exp\left(\gamma L_1\right)\frac{\gamma L_1}{n}\sum_{t=0}^{k-1}(1-\eta)^{k-t}\sum_{i=1}^n \mathrm{E}\left[\left\|\nabla f_i(x^t)\right\|\right].
\end{aligned}
$$

*In addition, for any $k \geq 0$,*

$$
\begin{aligned}
\frac{1}{n}\sum_{i=1}^n \mathrm{E}\left[\left\|v_i^k - \nabla f_i(x^k)\right\|\right] &\leq \frac{(1-\eta)^k}{n}\sum_{i=1}^n \mathrm{E}\left[\left\|v_i^0 - \nabla f_i(x^0)\right\|\right] + \sqrt{\eta}\sigma + \frac{\gamma}{\eta}L_0\exp\left(\gamma L_1\right) \\
&\quad + \exp\left(\gamma L_1\right)\frac{\gamma L_1}{n}\sum_{t=0}^{k}(1-\eta)^{k-t}\sum_{i=1}^n \mathrm{E}\left[\left\|\nabla f_i(x^t)\right\|\right],
\end{aligned}
$$

*Proof.* We prove the result using proof arguments similar to those of Theorem 1 in Cutkosky & Mehta (2020). From the definition of $v_i^{k+1}$, we have the following recursion for any $k \geq 0$:

$$
\begin{aligned}
v_i^{k+1} &= (1-\eta_{k+1})v_i^k + \eta_{k+1}\nabla f_i(x^{k+1};\xi_i^{k+1}) \\
&= \nabla f_i(x^{k+1}) + (1-\eta_{k+1})(v_i^k - \nabla f_i(x^k)) + (1-\eta_{k+1})(\nabla f_i(x^k) - \nabla f_i(x^{k+1})) \\
&\quad + \eta_{k+1}(\nabla f_i(x^{k+1};\xi_i^{k+1}) - \nabla f_i(x^{k+1})).
\end{aligned}
$$

Next, from the recursion of $v_i^{k+1}$, we obtain the following recursion for $k \geq 0$:

$$
H_i^{k+1} = (1-\eta_{k+1})H_i^k + (1-\eta_{k+1})G_i^k + \eta_{k+1}U_i^{k+1},
$$

where

$$
U_i^{k+1} = \nabla f_i(x^{k+1};\xi_i^{k+1}) - \nabla f_i(x^{k+1}), \quad G_i^k = \nabla f_i(x^k) - \nabla f_i(x^{k+1}), \quad H_i^k = v_i^k - \nabla f_i(x^k),
$$

$$U^{k+1} = \frac{1}{n} \sum_{i=1}^{n} U_i^{k+1}, \quad G^k = \frac{1}{n} \sum_{i=1}^{n} G_i^k, \quad \text{and} \quad H^k = \frac{1}{n} \sum_{i=1}^{n} H_i^k.$$

By applying the recursion of $H_i^k$ recursively, and by the fact that $(1 - \eta_{t+1}) \prod_{j=t+1}^{k}(1 - \eta_{j+1}) = \prod_{j=t}^{k}(1 - \eta_{j+1})$,

$$H_i^{k+1} = \prod_{t=0}^{k}(1 - \eta_{t+1})H_i^0 + \sum_{t=0}^{k} \prod_{j=t+1}^{k}(1 - \eta_{j+1})(1 - \eta_{t+1})G_i^t + \sum_{t=0}^{k} \prod_{j=t+1}^{k}(1 - \eta_{j+1})\eta_{t+1}U_i^{t+1}$$

$$= \prod_{t=0}^{k}(1 - \eta_{t+1})H_i^0 + \sum_{t=0}^{k} \prod_{j=t}^{k}(1 - \eta_{j+1})G_i^t + \sum_{t=0}^{k} \prod_{j=t+1}^{k}(1 - \eta_{j+1})\eta_{t+1}U_i^{t+1}.$$

By the fact that $H^k = \frac{1}{n} \sum_{i=1}^{n} H_i^k$,

$$H^{k+1} = \prod_{t=0}^{k}(1 - \eta_{t+1})H^0 + \sum_{t=0}^{k} \prod_{j=t}^{k}(1 - \eta_{j+1})G^t + \sum_{t=0}^{k} \prod_{j=t+1}^{k}(1 - \eta_{j+1})\eta_{t+1}U^{t+1}.$$

Next, taking the Euclidean norm, using the triangle inequality, and then taking the expectation, we obtain

$$\mathrm{E}\left[\left\|H^{k+1}\right\|\right] \leq \prod_{t=0}^{k}(1 - \eta_{t+1})\mathrm{E}\left[\left\|H^0\right\|\right] + \underbrace{\sum_{t=0}^{k} \prod_{j=t}^{k}(1 - \eta_{j+1})\mathrm{E}\left[\left\|G^t\right\|\right]}_{:=\mathcal{A}_1}$$

$$+ \underbrace{\mathrm{E}\left[\left\|\sum_{t=0}^{k} \prod_{j=t+1}^{k}(1 - \eta_{j+1})\eta_{t+1}U^{t+1}\right\|\right]}_{:=\mathcal{A}_2}. \tag{12}$$

To bound $\mathrm{E}\left[\left\|H^{k+1}\right\|\right]$, we need to bound the expectation of the last two terms. First, we bound term $\mathcal{A}_1$. By the fact that $\|G^t\| \leq \frac{1}{n} \sum_{i=1}^{n} \|G_i^t\|$, and by the definition of $G_i^t$,

$$\mathcal{A}_1 \leq \frac{1}{n} \sum_{i=1}^{n} \sum_{t=0}^{k} \prod_{j=t}^{k}(1 - \eta_{j+1})\mathrm{E}\left[\left\|\nabla f_i(x^t) - \nabla f_i(x^{t+1})\right\|\right]$$

$$\overset{(6)}{\leq} \frac{1}{n} \sum_{i=1}^{n} \sum_{t=0}^{k} \prod_{j=t}^{k}(1 - \eta_{j+1})\mathrm{E}\left[L_0 \exp\left(L_1 \left\|x^{t+1} - x^t\right\|\right) \left\|x^{t+1} - x^t\right\|\right]$$

$$+ \frac{1}{n} \sum_{i=1}^{n} \sum_{t=0}^{k} \prod_{j=t}^{k}(1 - \eta_{j+1})\mathrm{E}\left[L_1 \left\|\nabla f_i(x^t)\right\| \exp\left(L_1 \left\|x^{t+1} - x^t\right\|\right) \left\|x^{t+1} - x^t\right\|\right]$$

$$= \sum_{t=0}^{k} \prod_{j=t}^{k}(1 - \eta_{j+1})\gamma_t \exp(\gamma_t L_1)L_0 + \frac{L_1}{n} \sum_{i=1}^{n} \sum_{t=0}^{k} \prod_{j=t}^{k}(1 - \eta_{j+1})\gamma_t \exp\left(\gamma_t L_1\right) \mathrm{E}\left[\left\|\nabla f_i(x^t)\right\|\right].$$

Second, we bound term $\mathcal{A}_2$. By the independence of each sample variable $\xi_i^t$,

$$\mathcal{A}_2 \leq \sqrt{\mathrm{E}\left[\left\|\sum_{t=0}^{k} \prod_{j=t+1}^{k}(1 - \eta_{j+1})\eta_{t+1}U^{t+1}\right\|^2\right]}$$

$$= \sqrt{\sum_{t=0}^{k} \prod_{j=t+1}^{k}(1 - \eta_{j+1})^2 \eta_{t+1}^2 \mathrm{E}\left[\left\|U^{t+1}\right\|^2\right]}$$

Next, by the variance decomposition, i.e., $\mathrm{E}\left[\left\|U^{t+1}\right\|^2\right] = \frac{1}{n}\sum_{i=1}^n \mathrm{E}\left[\left\|U_i^{t+1}\right\|^2\right] \overset{(4)}{\leq} \sigma^2/n$,

$$
\begin{aligned}
\mathcal{A}_2 &\leq \sqrt{\sum_{t=0}^k \prod_{j=t+1}^k (1-\eta_{j+1})^2 \eta_{t+1}^2 \frac{\sigma^2}{n}} \\
&= \frac{\sigma}{\sqrt{n}}\sqrt{\sum_{t=0}^k \prod_{j=t+1}^k (1-\eta_{j+1})^2 \eta_{t+1}^2}.
\end{aligned}
$$

Therefore, by plugging the upper-bounds for $\mathcal{A}_1$, and for $\mathcal{A}_2$ into (12), we obtain

$$
\begin{aligned}
\mathrm{E}\left[\left\|H^{k+1}\right\|\right] &\leq \prod_{t=0}^k (1-\eta_{t+1})\mathrm{E}\left[\left\|H^0\right\|\right] + \sum_{t=0}^k \prod_{j=t}^k (1-\eta_{j+1})\gamma_t L_0 \exp\left(\gamma_t L_1\right) \\
&\quad + \frac{L_1}{n}\sum_{i=1}^n \sum_{t=0}^k \prod_{j=t}^k (1-\eta_{j+1})\gamma_t \exp\left(\gamma_t L_1\right) \mathrm{E}\left[\left\|\nabla f_i(x^t)\right\|\right] \\
&\quad + \frac{\sigma}{\sqrt{n}}\sqrt{\sum_{t=0}^k \prod_{j=t+1}^k (1-\eta_{j+1})^2 \eta_{t+1}^2}.
\end{aligned}
$$

Similarly, by following the proof arguments for bounding $\mathrm{E}\left[\left\|H^{k+1}\right\|\right]$, we can show the following inequality:

$$
\begin{aligned}
\frac{1}{n}\sum_{i=1}^n \mathrm{E}\left[\left\|H_i^{k+1}\right\|\right] &\leq \prod_{t=0}^k (1-\eta_{t+1})\frac{1}{n}\sum_{i=1}^n \mathrm{E}\left[\left\|H_i^0\right\|\right] + \sum_{t=0}^k \prod_{j=t}^k (1-\eta_{j+1})\gamma_t L_0 \exp\left(\gamma_t L_1\right) \\
&\quad + \frac{L_1}{n}\sum_{i=1}^n \sum_{t=0}^k \prod_{j=t}^k (1-\eta_{j+1})\gamma_t \exp\left(\gamma_t L_1\right) \mathrm{E}\left[\left\|\nabla f_i(x^t)\right\|\right] \\
&\quad + \sigma\sqrt{\sum_{t=0}^k \prod_{j=t+1}^k (1-\eta_{j+1})^2 \eta_{t+1}^2}.
\end{aligned}
$$

We further simplify our bounds. Let $\gamma_k \equiv \gamma > 0$, and $\eta_k \equiv \eta \in (0,1)$. Then, by the fact that

$$
\begin{aligned}
\prod_{t=0}^k (1-\eta_{t+1}) &= (1-\eta)^{k+1} \\
\sum_{t=0}^k \prod_{j=t}^k (1-\eta_{j+1})\gamma_t &= \gamma\sum_{t=0}^k (1-\eta)^{k-t+1}, \quad \text{and} \\
\sum_{t=0}^k \prod_{j=t+1}^k (1-\eta_{j+1})^2 \eta_{t+1}^2 &= \eta^2\sum_{t=0}^k (1-\eta)^{2(k-t)},
\end{aligned}
$$

we have

$$
\begin{aligned}
\mathrm{E}\left[\left\|H^{k+1}\right\|\right] &\leq (1-\eta)^{k+1}\mathrm{E}\left[\left\|H^0\right\|\right] + \gamma L_0 \exp\left(\gamma L_1\right)\sum_{t=0}^k (1-\eta)^{k-t+1} \\
&\quad + \exp\left(\gamma L_1\right)\frac{\gamma L_1}{n}\sum_{i=1}^n (1-\eta)^{k-t+1}\mathrm{E}\left[\left\|\nabla f_i(x^t)\right\|\right] + \frac{\sigma\eta}{\sqrt{n}}\sqrt{\sum_{t=0}^k (1-\eta)^{2(k-t)}}.
\end{aligned}
$$

By the fact that

$$\sum_{t=0}^{k}(1-\eta)^{k-t+1} \leq \sum_{t=0}^{\infty}(1-\eta)^t = \frac{1}{1-(1-\eta)} = \frac{1}{\eta};$$

$$\sum_{t=0}^{k}(1-\eta)^{2(k-t)} \leq \sum_{t=0}^{\infty}(1-\eta)^{2t} = \frac{1}{1-(1-\eta)^2} = \frac{1}{\eta(2-\eta)} \leq \frac{1}{\eta},$$

we obtain

$$\mathrm{E}\left[\left\|H^{k+1}\right\|\right] \leq (1-\eta)^{k+1}\mathrm{E}\left[\left\|H^0\right\|\right] + \frac{\gamma}{\eta}L_0\exp\left(\gamma L_1\right) + \frac{\sigma\sqrt{\eta}}{\sqrt{n}}$$

$$+ \exp\left(\gamma L_1\right)\frac{\gamma L_1}{n}\sum_{i=1}^{n}(1-\eta)^{k-t+1}\mathrm{E}\left[\left\|\nabla f_i(x^t)\right\|\right].$$

Similarly, by following the proof arguments for simplifying the bounds for $\mathrm{E}\left[\left\|H^{k+1}\right\|\right]$, we can show the following inequality:

$$\frac{1}{n}\sum_{i=1}^{n}\mathrm{E}\left[\left\|H^{k+1}\right\|\right] \leq \frac{(1-\eta)^{k+1}}{n}\sum_{i=1}^{n}\mathrm{E}\left[\left\|H_i^0\right\|\right] + \frac{\gamma}{\eta}L_0\exp\left(\gamma L_1\right) + \frac{\sigma\sqrt{\eta}}{\sqrt{n}}$$

$$+ \exp\left(\gamma L_1\right)\frac{\gamma L_1}{n}\sum_{i=1}^{n}(1-\eta)^{k-t+1}\mathrm{E}\left[\left\|\nabla f_i(x^t)\right\|\right].$$

$\square$

**Lemma 9.** *Let non-negative sequences $\{r^k\}$ and $\{s^k\}$ satisfy the following recursion: for $k = 0, 1, \ldots, K$, and $K \geq 0$,*

$$r^{k+1} \leq r^k - \gamma s^k + (1-\eta)^k\gamma a_1 + \gamma a_2 + \gamma^2 a_3\sum_{t=0}^{k}(1-\eta)^{k-t}r^t, \tag{13}$$

*where $a_1, a_2, a_3 > 0$, $\gamma > 0$, $\eta \in (0,1]$. If $\gamma^2/\eta a_3(K+1) \leq 1/2$, then for $k = 0, 1, \ldots, K$, and $K \geq 0$,*

$$r^k \leq p^k r^0 + ke,$$

*where $p$ and $e$ are defined by*

$$p = 1 + \frac{\gamma^2}{\eta}a_3, \quad and \quad e = \frac{\gamma(a_1 + a_2)}{1 - \gamma^2/\eta a_3(K+1)}.$$

*In addition, for $K \geq 0$,*

$$\min_{0 \leq k \leq K} s^k \leq \frac{2r^0}{\gamma(K+1)} + \frac{a_1}{\eta(K+1)} + \frac{3}{2}a_2 + \frac{1}{2}a_1.$$

*Proof.* We prove two statements in this lemma.

**Deriving the recursion of $r^k$ satisfying (13).** First, we prove that $r^k \leq p^k r^0 + ke$ satisfies the recursion in (13) by an induction. For $k = 0$, $r^0 \leq r^0$. Next, if $r^k \leq p^k r^0 + ke$ holds for $k$, then we prove this recursion for $k+1$:

$$r^{k+1} \leq r^k - \gamma s^k + (1-\eta)^k\gamma a_1 + \gamma a_2 + \gamma^2 a_3\sum_{t=0}^{k}(1-\eta)^{k-t}r^t$$

$$\leq p^k r_0 + ke + (1-\eta)^k\gamma a_1 + \gamma a_2 + \gamma^2 a_3\sum_{t=0}^{k}(1-\eta)^{k-t}(p^t r_0 + te).$$

Since

$$\sum_{t=0}^{k}(1-\eta)^{k-t}p^t r_0 \;\leq\; p^k r_0 \sum_{t=0}^{\infty}(1-\eta)^t = \frac{p^k r_0}{\eta}, \quad \text{and}$$

$$\sum_{t=0}^{k}(1-\eta)^{k-t}te \;\leq\; ke\sum_{t=0}^{\infty}(1-\eta)^t \leq \frac{ke}{\eta},$$

we obtain

$$r^{k+1} \;\leq\; p^k r_0 + ke + (1-\eta)^k \gamma a_1 + \gamma a_2 + \gamma^2 a_3 \frac{p^k r_0}{\eta} + \gamma^2 a_3 \frac{ke}{\eta}.$$

By re-arranging the terms, by the fact that $(1-\eta)^k \leq 1$, and by the fact that $k \leq K$,

$$r^{k+1} \;\leq\; \left(1 + \frac{\gamma^2}{\eta}a_3\right)p^k r_0 + ke + \gamma(a_1 + a_2) + e\frac{\gamma^2 a_3 K}{\eta}.$$

If $p = 1 + \frac{\gamma^2}{\eta}a_3$, $e = \frac{\gamma(a_1+a_2)}{1-\gamma^2/\eta a_3(K+1)}$, and $\gamma^2/\eta a_3(K+1) \leq 1/2$, then we can show that $\gamma(a_1 + a_2) + e\frac{\gamma^2 a_3 K}{\eta} = e$, and that

$$r^{k+1} \leq p^{k+1}r_0 + (k+1)e.$$

Thus, we complete the proof for the first statement.

**Deriving the convergence bound in** $\min_{0\leq k\leq K} s^k$**.** Next, based the derived inequality $r^k \leq p^k r_0 + ke$, we prove the second statement: the convergence in $\min_{0\leq k\leq K} s^k$. By summing (13) over $k = 0, 1, \ldots, K$,

$$\gamma\sum_{k=0}^{K}s^k \;\leq\; \sum_{k=0}^{K}(r^k - r^{k+1}) + \sum_{k=0}^{K}(1-\eta)^k\gamma a_1 + \gamma a_2(K+1) + \gamma^2 a_3\sum_{k=0}^{K}\sum_{t=0}^{k}(1-\eta)^{k-t}r^t$$

$$\leq\; r_0 + \frac{\gamma}{\eta}a_1 + \gamma a_2(K+1) + \gamma^2 a_3\sum_{k=0}^{K}\sum_{t=0}^{k}(1-\eta)^{k-t}r^t, \tag{14}$$

where we reach the last inequality by the fact that $r^{K+1} \geq 0$, and that $\sum_{k=0}^{K}(1-\eta)^k \leq \sum_{k=0}^{\infty}(1-\eta)^k = 1/\eta$. To complete the convergence bound, we need to bound the last term from the previous inequality:

$$\sum_{k=0}^{K}\sum_{t=0}^{k}(1-\eta)^{k-t}r^t \;=\; \sum_{t=0}^{K}\sum_{k=t}^{K}(1-\eta)^{k-t}r^t$$

$$=\; \sum_{t=0}^{K}\frac{r^t}{(1-\eta)^t}\sum_{k=t}^{K}(1-\eta)^k$$

$$=\; \sum_{t=0}^{K}\frac{r^t}{(1-\eta)^t}\cdot(1-\eta)^t\frac{1-(1-\eta)^{K-t}}{1-(1-\eta)}$$

$$\leq\; \frac{1}{\eta}\sum_{k=0}^{K}r^k.$$

By the inequality $r^k \leq p^k r_0 + ke$,

$$\sum_{k=0}^{K}\sum_{t=0}^{k}(1-\eta)^{k-t}r^t \;\leq\; \frac{1}{\eta}\sum_{k=0}^{K}(p^k r_0 + ke) = \frac{1}{\eta}\left(\frac{p^{K+1}-1}{p-1}r_0 + \frac{K(K+1)}{2}e\right).$$

Plugging the upper-bound for $\sum_{k=0}^{K}\sum_{t=0}^{k}(1-\eta)^{k-t}r^t$ into (14) yields

$$\gamma\sum_{k=0}^{K}s^k \;\leq\; r_0 + \frac{\gamma}{\eta}a_1 + \gamma a_2(K+1) + \frac{\gamma^2}{\eta}a_3\frac{p^{K+1}-1}{p-1}r_0 + \frac{\gamma^2}{\eta}a_3\frac{K(K+1)}{2}e.$$

Next, by the fact that $p = 1 + \frac{\gamma^2}{\eta} a_3$ and $e = \frac{\gamma(a_1+a_2)}{1-\gamma^2/\eta a_3(K+1)}$,

$$
\begin{aligned}
\gamma \sum_{k=0}^{K} s^k &\leq r_0 + \left(1 + \frac{\gamma^2}{\eta} a_3\right)^{K+1} r_0 + \frac{\gamma}{\eta} a_1 + \gamma a_2 (K+1) + \frac{\gamma^2}{\eta} a_3 \frac{K(K+1)}{2} \frac{\gamma(a_1+a_2)}{1-\gamma^2/\eta a_3(K+1)} \\
&\leq r_0 + \exp\left(\frac{\gamma^2}{\eta} a_3 (K+1)\right) r_0 + \frac{\gamma}{\eta} a_1 + \gamma a_2 (K+1) + \frac{K}{2} \frac{\gamma^2(K+1)}{\eta} a_3 \frac{\gamma(a_1+a_2)}{1-\gamma^2/\eta a_3(K+1)}.
\end{aligned}
$$

By the fact that $\frac{\gamma^2(K+1)}{\eta} a_3 \leq \frac{1}{2}$,

$$
\gamma \sum_{k=0}^{K} s^k \leq 2r_0 + \frac{\gamma}{\eta} a_1 + \gamma a_2 (K+1) + \frac{K}{2} \gamma(a_1+a_2)
$$

Finally, using that $\gamma \sum\limits_{k=0}^{K} s^k \geq \gamma(K+1) \min\limits_{0 \leq k \leq K} s^k$, we obtain

$$
\gamma(K+1) \min_{0 \leq k \leq K} s^k \leq 2r_0 + \frac{\gamma}{\eta} a_1 + \gamma a_2 (K+1) + \frac{K}{2} \gamma(a_1+a_2),
$$

which completes the proof. $\qquad\square$

### D.2 PROOF OF THEOREM 2

Now, we are ready to prove Theorem 2. First of all, define the Lyaponov function $V_k$ for any $k \geq 0$

$$
V_k = f(x^k) - f^{\inf} + \frac{A}{n} \sum_{i=1}^{n} \left\| v_i^{k+1} - g_i^{k+1} \right\|,
$$

with $A = \frac{2\gamma}{1-\sqrt{1-\alpha}}$. By Lemma 6 and 7,

$$
\begin{aligned}
\mathrm{E}\left[V_{k+1}\right] &\leq \mathrm{E}\left[f(x^k) - f^{\inf}\right] - \gamma \mathrm{E}\left[\left\|\nabla f(x^k)\right\|\right] + 2\gamma \mathrm{E}\left[\left\|\nabla f(x^k) - v^k\right\|\right] + 2\gamma \mathrm{E}\left[\left\|v^k - g^k\right\|\right] \\
&\quad + A\sqrt{1-\alpha} \frac{1}{n} \sum_{i=1}^{n} \mathrm{E}\left[\left\|v_i^k - g_i^k\right\|\right] + A\sqrt{1-\alpha} \frac{\eta}{n} \sum_{i=1}^{n} \mathrm{E}\left[\left\|v_i^k - \nabla f_i(x^k)\right\|\right] \\
&\quad + \frac{\gamma^2}{2} \exp\left(\gamma L_1\right) \left(L_0 + \frac{L_1}{n} \sum_{i=1}^{n} \mathrm{E}\left[\left\|\nabla f_i(x^k)\right\|\right]\right) \\
&\quad + A\sqrt{1-\alpha} \eta\gamma \exp\left(\gamma L_1\right) \left(L_0 + \frac{L_1}{n} \sum_{i=1}^{n} \mathrm{E}\left[\left\|\nabla f_i(x^k)\right\|\right]\right) + A\sqrt{1-\alpha} \eta\sigma.
\end{aligned}
$$

Since $A = \frac{2\gamma}{1-\sqrt{1-\alpha}}$, we obtain $A\sqrt{1-\alpha}\eta = 2\gamma\eta C_\alpha$, where $C_\alpha = \frac{\sqrt{1-\alpha}}{1-\sqrt{1-\alpha}}$, and

$$
\begin{aligned}
\mathrm{E}\left[V_{k+1}\right] &\leq \mathrm{E}\left[V_k\right] - \gamma \mathrm{E}\left[\left\|\nabla f(x^k)\right\|\right] + 2\gamma \mathrm{E}\left[\left\|\nabla f(x^k) - v^k\right\|\right] + 2\gamma\eta \frac{C_\alpha}{n} \sum_{i=1}^{n} \mathrm{E}\left[\left\|v_i^k - \nabla f_i(x^k)\right\|\right] \\
&\quad + \gamma^2 \left(\frac{1}{2} + 2C_\alpha\eta\right) \exp\left(\gamma L_1\right) \left(L_0 + \frac{L_1}{n} \sum_{i=1}^{n} \mathrm{E}\left[\left\|\nabla f_i(x^k)\right\|\right]\right) + 2\gamma\eta C_\alpha\sigma.
\end{aligned}
$$

By Lemma 8,

$$
\begin{aligned}
\mathrm{E}\left[V_{k+1}\right] \leq{} & \mathrm{E}\left[V_k\right] - \gamma \mathrm{E}\left[\left\|\nabla f(x^k)\right\|\right] + 2\gamma(1-\eta)^k \mathrm{E}\left[\left\|v^0 - \nabla f(x^0)\right\|\right] + \frac{2\gamma\sqrt{\eta}\sigma}{n} \\
& + \frac{2\gamma^2}{\eta} L_0 \exp\left(\gamma L_1\right) + \frac{2\gamma^2 L_1}{n} \sum_{t=0}^{k-1}(1-\eta)^{k-t} \sum_{i=1}^{n} \mathrm{E}\left[\left\|\nabla f_i(x^t)\right\|\right] \exp\left(\gamma L_1\right) \\
& + 2\gamma\eta C_\alpha \left(\frac{(1-\eta)^k}{n} \sum_{i=1}^{n} \mathrm{E}\left[\left\|v_i^0 - \nabla f_i(x^0)\right\|\right] + \sqrt{\eta}\sigma + \frac{\gamma}{\eta} L_0 \exp\left(\gamma L_1\right)\right) \\
& + 2\gamma\eta C_\alpha \exp\left(\gamma L_1\right) \cdot \frac{\gamma L_1}{n} \sum_{t=0}^{k}(1-\eta)^{k-t} \sum_{i=1}^{n} \mathrm{E}\left[\left\|\nabla f_i(x^t)\right\|\right] \\
& + \gamma^2 \left(\frac{1}{2} + 2C_\alpha\eta\right) \exp\left(\gamma L_1\right) \left(L_0 + \frac{L_1}{n} \sum_{i=1}^{n} \mathrm{E}\left[\left\|\nabla f_i(x^k)\right\|\right]\right) + 2\gamma\eta C_\alpha\sigma.
\end{aligned}
$$

Denoting $\mathcal{V}_0 = \left\|v^0 - \nabla f(x^0)\right\|$ and $\widetilde{\mathcal{V}}_0 = \frac{1}{n}\sum_{i=1}^{n}\left\|v_i^0 - \nabla f_i(x^0)\right\|$, we have

$$
\begin{aligned}
\mathrm{E}\left[V_{k+1}\right] \leq{} & \mathrm{E}\left[V_k\right] - \gamma \mathrm{E}\left[\left\|\nabla f(x^k)\right\|\right] + (1-\eta)^k \left(2\gamma \mathrm{E}\left[\mathcal{V}_0\right] + 2\gamma\eta C_\alpha \mathrm{E}\left[\widetilde{\mathcal{V}}_0\right]\right) \\
& + 2\gamma\left(\sqrt{\frac{\eta}{n}} + \eta^{3/2} C_\alpha + \eta C_\alpha\right)\sigma + \gamma\left(\frac{2\gamma}{\eta} + 2\gamma C_\alpha + \frac{\gamma}{2} + 2\gamma\eta C_\alpha\right) L_0 \exp\left(\gamma L_1\right) \\
& + 2\gamma^2 \left(1 + \eta C_\alpha\right) \exp\left(\gamma L_1\right) \frac{L_1}{n} \sum_{i=1}^{n} \sum_{t=0}^{k-1}(1-\eta)^{k-t} \mathrm{E}\left[\left\|\nabla f_i(x^t)\right\|\right] \\
& + \gamma^2 \left(\frac{1}{2} + 2\eta C_\alpha\right) \exp\left(\gamma L_1\right) \frac{L_1}{n} \sum_{i=1}^{n} \mathrm{E}\left[\left\|\nabla f_i(x^k)\right\|\right].
\end{aligned}
$$

Applying Lemma 2, we obtain

$$
\begin{aligned}
\mathrm{E}\left[V_{k+1}\right] \leq{} & \mathrm{E}\left[V_k\right] - \gamma \mathrm{E}\left[\left\|\nabla f(x^k)\right\|\right] + (1-\eta)^k \left(2\gamma \mathrm{E}\left[\mathcal{V}_0\right] + 2\gamma\eta C_\alpha \mathrm{E}\left[\widetilde{\mathcal{V}}_0\right]\right) \\
& + 2\gamma\left(\sqrt{\frac{\eta}{n}} + \eta^{3/2} C_\alpha + \eta C_\alpha\right)\sigma + \gamma\left(\frac{2\gamma}{\eta} + 2\gamma C_\alpha + \frac{\gamma}{2} + 2\gamma\eta C_\alpha\right) L_0 \exp\left(\gamma L_1\right) \\
& + 2\gamma^2 \exp\left(\gamma L_1\right)\left(1 + \eta C_\alpha\right) \sum_{t=0}^{k-1}(1-\eta)^{k-t}\left(8L_1^2 \mathrm{E}\left[f(x^t) - f^{\inf}\right] + \frac{8L_1^2}{n}\sum_{i=1}^{n}(f^{\inf} - f_i^{\inf}) + L_0\right) \\
& + \gamma^2 \exp\left(\gamma L_1\right)\left(\frac{1}{2} + 2\eta C_\alpha\right)\left(8L_1^2 \mathrm{E}\left[f(x^k) - f^{\inf}\right] + \frac{8L_1^2}{n}\sum_{i=1}^{n}(f^{\inf} - f_i^{\inf}) + L_0\right).
\end{aligned}
$$

By re-arranging the terms,

$$
\begin{aligned}
\mathrm{E}\left[V_{k+1}\right] \leq{} & \mathrm{E}\left[V_k\right] - \gamma \mathrm{E}\left[\left\|\nabla f(x^k)\right\|\right] + (1-\eta)^k \left(2\gamma \mathrm{E}\left[\mathcal{V}_0\right] + 2\gamma\eta C_\alpha \mathrm{E}\left[\widetilde{\mathcal{V}}_0\right]\right) \\
& + 16\gamma^2(1 + \eta C_\alpha) \exp\left(\gamma L_1\right) L_1^2 \sum_{t=0}^{k}(1-\eta)^{k-t} \mathrm{E}\left[f(x^t) - f^{\inf}\right] \\
& + 2\gamma\left(\sqrt{\frac{\eta}{n}} + \eta^{3/2} C_\alpha + \eta C_\alpha\right)\sigma + \gamma\left(\frac{2\gamma}{\eta} + 2\gamma C_\alpha + \frac{\gamma}{2} + 2\gamma\eta C_\alpha\right) \exp\left(\gamma L_1\right) L_0 \\
& + \left(\frac{\gamma^2}{2} + 2\gamma^2\eta C_\alpha + 2\gamma^2(1 + \eta C_\alpha) \sum_{t=0}^{k-1}(1-\eta)^{k-t}\right) \exp\left(\gamma L_1\right) L_0 \\
& + 8\left(\frac{\gamma^2}{2} + 2\gamma^2\eta C_\alpha + 2\gamma^2(1 + \eta C_\alpha) \sum_{t=0}^{k-1}(1-\eta)^{k-t}\right) \exp\left(\gamma L_1\right) \frac{L_1^2}{n}\sum_{i=1}^{n}(f^{\inf} - f_i^{\inf}).
\end{aligned}
$$

Next, by the fact that $\sum_{t=0}^{k-1}(1-\eta)^{k-t} \leq \sum_{t=0}^{\infty}(1-\eta) = \frac{1}{\eta}$,

$$2\gamma^2(1+\eta C_\alpha)\sum_{t=0}^{k-1}(1-\eta)^{k-t} \leq \frac{2\gamma^2 + 2\gamma^2\eta C_\alpha}{\eta}$$

$$= \frac{2\gamma^2}{\eta} + 2\gamma^2 C_\alpha.$$

Therefore, by using the upper bound of $2\gamma^2(1+\eta C_\alpha)\sum_{t=0}^{k-1}(1-\eta)^{k-t}$, and by the fact that $f(x^t) - f^{\inf} \leq V_t$,

$$\mathrm{E}\left[V_{k+1}\right] \leq \mathrm{E}\left[V_k\right] - \gamma\mathrm{E}\left[\left\|\nabla f(x^k)\right\|\right] + (1-\eta)^k\left(2\gamma\mathrm{E}\left[\mathcal{V}_0\right] + 2\gamma\eta C_\alpha\mathrm{E}\left[\widetilde{\mathcal{V}}_0\right]\right)$$

$$+ 16\gamma^2(1+\eta C_\alpha)\exp\left(\gamma L_1\right)L_1^2\sum_{t=0}^{k}(1-\eta)^{k-t}\mathrm{E}\left[V_t\right] + 2\gamma\left(\sqrt{\frac{\eta}{n}} + \eta(1+\sqrt{\eta})C_\alpha\right)\sigma$$

$$+ \gamma\exp\left(\gamma L_1\right)L_0\left(\gamma + \frac{4\gamma}{\eta} + 4\gamma(1+\eta)C_\alpha\right)$$

$$+ 4\gamma\exp\left(\gamma L_1\right)\left(\gamma + \frac{4\gamma}{\eta} + 4\gamma(1+\eta)C_\alpha\right)\frac{L_1^2}{n}\sum_{i=1}^{n}\left(f^{\inf} - f_i^{\inf}\right).$$

By assuming that $16\gamma^2/\eta(K+1)(1+\eta C_\alpha)L_1^2\exp\left(\gamma L_1\right) \leq \frac{1}{2}$, and applying Lemma 9 with $s^k = \mathrm{E}\left[\left\|\nabla f(x^k)\right\|\right]$, $r^k = \mathrm{E}\left[V_k\right]$, $a_1 = 2\mathrm{E}\left[\mathcal{V}_0\right] + 2\eta C_\alpha\mathrm{E}\left[\widetilde{\mathcal{V}}_0\right]$, $a_2 = 2\left(\sqrt{\frac{\eta}{n}} + \eta(1+\sqrt{\eta})C_\alpha\right)\sigma + \exp\left(\gamma L_1\right)L_0\left(\gamma + \frac{4\gamma}{\eta} + 4\gamma(1+\eta)C_\alpha\right) + 4\exp\left(\gamma L_1\right)\left(\gamma + \frac{4\gamma}{\eta} + 4\gamma(1+\eta)C_\alpha\right)\frac{L_1^2}{n}\sum_{i=1}^{n}\left(f^{\inf} - f_i^{\inf}\right)$, and $a_3 = 16(1+\eta C_\alpha)\exp\left(\gamma L_1\right)L_1^2$, we get

$$\min_{0\leq k\leq K}\mathrm{E}\left[\left\|\nabla f(x^k)\right\|\right] \leq \frac{2\mathrm{E}\left[V_0\right]}{\gamma(K+1)} + \frac{2\mathrm{E}\left[\mathcal{V}_0\right] + \eta C_\alpha\mathrm{E}\left[\widetilde{\mathcal{V}}_0\right]}{\eta(K+1)} + \mathrm{E}\left[\mathcal{V}_0\right] + \eta C_\alpha\mathrm{E}\left[\widetilde{\mathcal{V}}_0\right]$$

$$+ 3\left(\sqrt{\frac{\eta}{n}} + \eta(1+\sqrt{\eta})C_\alpha\right)\sigma$$

$$+ \frac{3}{2}\exp\left(\gamma L_1\right)L_0\left(\gamma + \frac{4\gamma}{\eta} + 4\gamma(1+\eta)C_\alpha\right)$$

$$+ 6\exp\left(\gamma L_1\right)\left(\gamma + \frac{4\gamma}{\eta} + 4\gamma(1+\eta)C_\alpha\right)\frac{L_1^2}{n}\sum_{i=1}^{n}\left(f^{\inf} - f_i^{\inf}\right).$$

If $\eta = \frac{1}{\sqrt{K+1}}$, and $\gamma = \frac{\gamma_0}{(K+1)^{3/4}}$ with $\gamma_0 > 0$ satisfying

$$32\gamma_0^2 L_1^2\left(1 + \frac{C_\alpha}{\sqrt{K+1}}\right)\exp\left(\frac{\gamma_0 L_1}{(K+1)^{3/4}}\right) \leq 1,$$

then we have $\exp\left(\gamma L_1\right) = \exp\left(\frac{\gamma_0 L_1}{(K+1)^{3/4}}\right) \leq \exp\left(\gamma_0 L_1\right)$, and

$$\min_{0\leq k\leq K}\mathrm{E}\left[\left\|\nabla f(x^k)\right\|\right] \leq \frac{2\mathrm{E}\left[V_0\right]}{\gamma_0(K+1)^{1/4}} + \frac{2\mathrm{E}\left[\mathcal{V}_0\right] + \eta C_\alpha\mathrm{E}\left[\widetilde{\mathcal{V}}_0\right]}{(K+1)^{1/2}} + \mathrm{E}\left[\mathcal{V}_0\right] + \frac{C_\alpha\mathrm{E}\left[\widetilde{\mathcal{V}}_0\right]}{(K+1)^{1/2}}$$

$$+ 3\left(\frac{1}{\sqrt{n}(K+1)^{1/4}} + \frac{2C_\alpha}{(K+1)^{1/2}}\right)\sigma$$

$$+ \frac{3}{2}\exp\left(\gamma_0 L_1\right)L_0\left(\frac{\gamma_0}{(K+1)^{3/4}} + \frac{4\gamma_0}{(K+1)^{1/4}} + \frac{8\gamma_0 C_\alpha}{(K+1)^{3/4}}\right)$$

$$+ 6\exp\left(\gamma_0 L_1\right)\left(\frac{\gamma_0}{(K+1)^{3/4}} + \frac{4\gamma_0}{(K+1)^{1/4}} + \frac{8\gamma_0 C_\alpha}{(K+1)^{3/4}}\right)L_1^2\delta^{\inf},$$

where $\delta^{\inf} = \frac{1}{n} \sum_{i=1}^{n} \left( f^{\inf} - f_i^{\inf} \right)$. By the fact that $C_\alpha = \frac{\sqrt{1-\alpha}(1+\sqrt{1-\alpha})}{\alpha} \leq \frac{2\sqrt{1-\alpha}}{\alpha}$,

$$\min_{0 \leq k \leq K} \mathrm{E}\left[ \|\nabla f(x^k)\| \right] \leq \mathcal{O}\left( \frac{\mathrm{E}[\mathcal{V}_0]/\gamma_0 + \sigma/\sqrt{n} + (\gamma_0 L_0 + \gamma_0 L_1^2 \delta^{\inf}) \exp(\gamma_0 L_1)}{(K+1)^{1/4}} + \mathrm{E}\left[\mathcal{V}_0\right] \right)$$
$$+ \mathcal{O}\left( \frac{\sqrt{1-\alpha}}{\alpha} \cdot \frac{\mathrm{E}\left[\widetilde{\mathcal{V}_0}\right] + \sigma + (L_0\gamma_0 + \gamma_0 L_1^2 \delta^{\inf}) \exp(\gamma_0 L_1)}{(K+1)^{1/2}} \right).$$

If $v_i^0$ is initialized to be the mini-batch stochastic gradient at the starting point with batch size $B^{\mathrm{init}} \in [n]$:

$$v_i^0 = \frac{1}{B^{\mathrm{init}}} \sum_{j=1}^{B^{\mathrm{init}}} \nabla f_i(x_i^0; \xi_{i,j}^0),$$

where $\xi_{i,j}^0$ are i.i.d., $j \in B^{\mathrm{init}}$, then we have the following bounds for $\mathrm{E}\left[\mathcal{V}_0\right]$, and $\mathrm{E}\left[\widetilde{\mathcal{V}_0}\right]$:

$$\mathrm{E}\left[\mathcal{V}_0\right] = \mathrm{E}\left[ \left\| \frac{1}{nB^{\mathrm{init}}} \sum_i^n \sum_{j=1}^{B^{\mathrm{init}}} \nabla f_i(x_i^0; \xi_{i,j}^0) - \nabla f(x^0) \right\| \right]$$
$$\leq \sqrt{\mathrm{E}\left[ \left\| \frac{1}{nB^{\mathrm{init}}} \sum_i^n \sum_{j=1}^{B^{\mathrm{init}}} \nabla f_i(x_i^0; \xi_{i,j}^0) - \nabla f(x^0) \right\|^2 \right]}$$
$$\leq \frac{\sigma}{\sqrt{nB^{\mathrm{init}}}}; \quad \text{and}$$
$$\mathrm{E}\left[\widetilde{\mathcal{V}_0}\right] = \frac{1}{n} \sum_{i=1}^{n} \mathrm{E}\left[ \left\| \frac{1}{B^{\mathrm{init}}} \sum_{j=1}^{B^{\mathrm{init}}} \nabla f_i(x_i^0; \xi_{i,j}^0) - \nabla f_i(x^0) \right\| \right]$$
$$\leq \frac{1}{n} \sum_{i=1}^{n} \sqrt{\mathrm{E}\left[ \left\| \frac{1}{B^{\mathrm{init}}} \sum_{j=1}^{B^{\mathrm{init}}} \nabla f_i(x_i^0; \xi_{i,j}^0) - \nabla f_i(x^0) \right\|^2 \right]}$$
$$\leq \frac{\sigma}{\sqrt{B^{\mathrm{init}}}}.$$

By taking $B^{\mathrm{init}} = \sqrt{K+1}$,

$$\mathrm{E}\left[\mathcal{V}_0\right] \leq \frac{\sigma}{\sqrt{n}(K+1)^{1/2}}, \quad \mathrm{E}\left[\widetilde{\mathcal{V}_0}\right] \leq \frac{\sigma}{(K+1)^{1/2}}.$$

Therefore,

$$\min_{0 \leq k \leq K} \mathrm{E}\left[ \|\nabla f(x^k)\| \right] \leq \mathcal{O}\left( \frac{\mathrm{E}[\mathcal{V}_0]/\gamma_0 + \sigma/\sqrt{n} + (\gamma_0 L_0 + \gamma_0 L_1^2 \delta^{\inf}) \exp(\gamma_0 L_1)}{(K+1)^{1/4}} \right)$$
$$+ \mathcal{O}\left( \frac{\sqrt{1-\alpha}}{\alpha} \cdot \frac{\sigma + (L_0\gamma_0 + \gamma_0 L_1^2 \delta^{\inf}) \exp(\gamma_0 L_1)}{(K+1)^{1/2}} \right).$$

# E  ADDITIONAL EXPERIMENTAL RESULTS

In this section, we provide additional results for minimizing nonconvex polynomial functions, and for training the ResNet-20 model over the CIFAR-10 dataset.

## E.1  MINIMIZATION OF NONCONVEX POLYNOMIAL FUNCTIONS

We ran normalized EF21 (`EF21-norm`), and traditional EF21 in a single-node setting ($n = 1$) for solving the following problem:

$$\min_{x \in \mathbb{R}^d} \left\{ f(x) := \underbrace{\sum_{i=1}^d a_i x_i^4}_{=:g(x)} + \lambda \underbrace{\sum_{i=1}^d \frac{x_i^2}{1 + x_i^2}}_{=:h(x)} \right\}, \tag{15}$$

where $a_i > 0$, $i = 1, \dots, d$, $\lambda > 0$.

Let us show that $f(x)$ is non-convex (for the specific choice of $a_i$) and $(L_0, L_1)$-smooth. First, we prove that $f(x)$ is non-convex. Indeed,

$$\nabla^2 f(x) = \nabla^2 g(x) + \nabla^2 h(x)$$

$$= 12 \operatorname{diag} \left\{ a_1 x_1^2, \dots, a_d x_d^2 \right\} + 2\lambda \operatorname{diag} \left\{ \frac{1 - 3x_1^2}{(1 + x_1^2)^3}, \dots, \frac{1 - 3x_d^2}{(1 + x_d^2)^3} \right\},$$

is not positive definite matrix if we choose $a_i = \frac{\lambda}{24}$, $x_i = \pm 1$ for $i = 1, \dots, d$.

Second, we find $L_0, L_1 > 0$ such that $\left\| \nabla^2 f(x) \right\| \le L_0 + L_1 \left\| \nabla f(x) \right\|$, $\forall x \in \mathbb{R}^d$. This condition is equivalent to Assumption 3 (generalized smoothness) with $L_0, L_1$ (Chen et al., 2023, Theorem 1). Let us fix some $L_1 > 0$ and choose $L_0 = \frac{9\lambda d^2}{2L_1^2} + 2\lambda$. Since $\nabla^2 h(x) \preccurlyeq 2\lambda I$,

$$\left\| \nabla^2 f(x) \right\| = \left\| \nabla^2 g(x) + \nabla^2 h(x) \right\| \le \left\| \nabla^2 g(x) \right\| + \left\| \nabla^2 h(x) \right\|$$

$$\le 12 \sqrt{a_1^2 x_1^4 + \dots + a_d^2 x_d^4} + 2\lambda$$

$$\le 12 \left( a_1 x_1^2 + \dots + a_d x_d^2 \right) + 2\lambda.$$

Also, notice that

$$\|\nabla f(x)\| = \|\nabla g(x) + \nabla h(x)\| = \sqrt{\left( 4a_1 x_1^2 + \frac{2\lambda}{(1 + x_1^2)^2} \right)^2 x_1^2 + \dots + \left( 4a_d x_d^2 + \frac{2\lambda}{(1 + x_d^2)^2} \right)^2 x_d^2}$$

$$\ge 4 \sqrt{a_1^2 x_1^6 + \dots + a_d^2 x_d^6}$$

$$\overset{(*)}{\ge} \frac{4}{\sqrt{d}} \left( a_1 |x_1|^3 + \dots + a_d |x_d|^3 \right),$$

where (*) results from the fact that $\|x\|_1 \le \sqrt{d} \|x\|$ for $x \in \mathbb{R}^d$. Our goal is to show that

$$12 \left( a_1 x_1^2 + \dots + a_d x_d^2 \right) \le \tilde{L}_0 + \frac{4L_1}{\sqrt{d}} \left( a_1 |x_1|^3 + \dots + a_d |x_d|^3 \right), \quad \tilde{L}_0 = L_0 - 2\lambda.$$

To show this, we consider two cases: if $|x_i| \le \frac{3\sqrt{d}}{L_1}$, and otherwise.

1. If $|x_i| \le \frac{3\sqrt{d}}{L_1}$ for all $i = 1, \dots, d$, then $12a_i x_i^2 \le \frac{108 a_i d}{L_1^2}$. Thus, $12 \left( a_1 x_1^2 + \dots + a_d x_d^2 \right) \le \frac{108 \lambda d^2}{24 L_1^2} = \tilde{L}_0$.

2. If $|x_j| > \frac{3\sqrt{d}}{L_1}$ for some $j = 1, \dots, d$, then $12a_j x_j^2 < \frac{4L_1}{\sqrt{d}} a_j |x_j|^3$, and the sum of the remaining terms (such that $|x_i| \le \frac{3\sqrt{d}}{L_1}$) in $12 \left( a_1 x_1^2 + \dots + a_d x_d^2 \right)$ can be upper bounded by $\tilde{L}_0$.

In conclusion, $f(x)$ is $(L_0, L_1)$-smooth, where $L_1$ is any positive constant and $L_0 = \frac{9\lambda d^2}{2L_1^2} + 2\lambda$.

Additionally, we can show that under certain additional constraints, $f(x)$ is $L$-smooth with $L = \frac{\lambda\sqrt{d}D^2}{2} + 2\lambda$. If $|x_i| \le D$ for all $i = 1, \ldots, d$, then

$$\left\| \nabla^2 f(x) \right\| \le 12\sqrt{a_1^2 x_1^4 + \ldots + a_d^2 x_d^4} + 2\lambda \le \frac{\lambda\sqrt{d}D^2}{2} + 2\lambda = L,$$

In the experiments, we estimate $D$ based on the initial point $x^0 \in \mathbb{R}^d$.

In the following experiments, we used a top-$k$ sparsifier with $k = 1$ and $\alpha = k/d$, setting $d = 4$, $L_1 = \{1, 4, 8\}$, and $L_0 = 4$ (adjusting $\lambda$ to maintain a constant $L_0$). The initial values $x^0$ were drawn from a normal distribution, $x_i^0 \sim \mathcal{N}(20, 1)$ for $i = 1, \ldots, d$, with $D$ estimated as 20. For EF21, we set $\gamma_k = \frac{1}{L + L\sqrt{\frac{\beta}{\theta}}}$, using $\theta = 1 - \sqrt{1-\alpha}$ and $\beta = \frac{1-\alpha}{1-\sqrt{1-\alpha}}$, according to Theorem 1 of Richtárik et al. (2021). For normalized EF21, we chose $\gamma_k = \frac{1}{2c_1}$ with $c_1 = \frac{L_1}{2} + 2\frac{\sqrt{1-\alpha}L_1}{1-\sqrt{1-\alpha}}$ from Theorem 3, and $\gamma_k = \frac{\gamma_0}{\sqrt{K+1}}$ with $\gamma_0 > 0$, as specified in Theorem 1 with $n = 1$.

**The impact of $\gamma_0$ and $K$ on the convergence of normalized EF21.** First, we investigate the impact of $\gamma_0$ and $K$ on the convergence of normalized EF21. We evaluated $\gamma_0$ from the set $\{0.1, 1, 10\}$, and plotted the histogram representing the number of iterations required to achieve the target accuracy of $\|\nabla f(x)\|^2 < \epsilon$ with $\epsilon = 10^{-4}$, using the stepsize rule $\gamma = \frac{\gamma_0}{\sqrt{K+1}}$. For each $\gamma_0$, we determined $K$ as the minimum number of iterations required to achieve the desired accuracy, found through a grid search with step sizes of 500 for $\gamma_0 = 1, 10$ and 5000 for $\gamma_0 = 0.1$. From Figure 4,

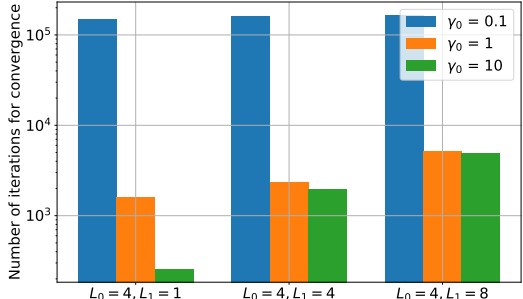

Figure 4: Number of iterations required to achieve the desired accuracy, $\|\nabla f(x)\|^2 < \epsilon$, $\epsilon = 10^{-4}$, using normalized EF21 (EF21-norm) with $\gamma = \frac{\gamma_0}{\sqrt{K+1}}$ for different values of $L_0$ and $L_1$.

for small values of $\gamma_0$, such as $0.1$, significantly more iterations are required to reach convergence compared to $\gamma_0$ values of 1 and 10, which show similar performance (with the exception of the $L_0 = 4, L_1 = 1$ case, where $\gamma_0 = 10$ converges faster). Based on this observation, we use $\gamma_0 = 1$ in all subsequent experiments and adjust only $K$ to achieve convergence, identifying the minimum number of iterations needed to reach the target accuracy through a grid search with a step size of 500.

**Comparisons between EF21 and normalized EF21.** Next, we evaluate the performance of EF21 and normalized EF21 for a fixed $L_0 = 4$ and varying $L_1$ values of $\{1, 4, 8\}$. From Figure 1, normalized EF21, regardless of the chosen stepsize $\gamma$, achieves the desired accuracy $\|\nabla f(x)\|^2 < \epsilon$ with $\epsilon = 10^{-4}$ faster than the original EF21. Initially, however, EF21 converges more quickly, likely because normalized EF21 employs normalized gradients, which can be slower at the start due to the large gradients when the initial point is far from the stationary point. Moreover, as $L_1$ increases, both methods show slower convergence.

### E.2 ResNet20 Training over CIFAR-10

We included additional experimental results from running EF21 and normalized EF21 for training the ResNet20 model over the CIFAR-10 dataset. The parameter details were set to be the same as

those in Section 6.2, with the exception that we vary $k = 0.01d, 0.5d$ for a top-$k$ sparsifier. From Figures 5 and 6, normalized EF21 attains a higher accuracy improvement than EF21, across different sparsification levels $k$.

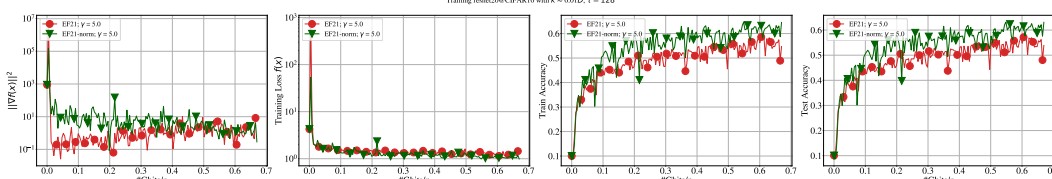

Figure 5: ResNet20 training on CIFAR-10 by using EF21 and normalized EF21 (`EF21-norm`) under the same stepsize $\gamma = 5$ and $k = 0.01d$ for a top-$k$ sparsifier.

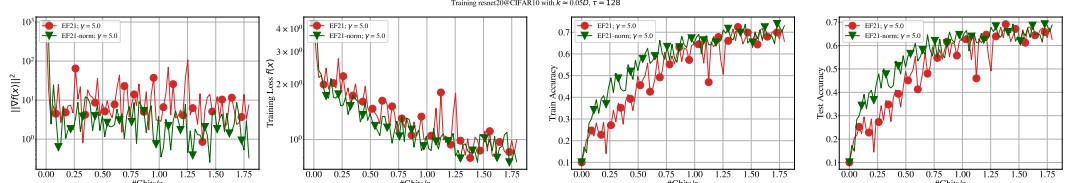

Figure 6: ResNet20 training on CIFAR-10 by using EF21 and normalized EF21 (`EF21-norm`) under the same stepsize $\gamma = 5$ and $k = 0.05d$ for a top-$k$ sparsifier.

# F    OMITTED PROOF FOR SMOOTHNESS PARAMETERS OF LOGISTIC REGRESSION

In this section, we prove the generalized smoothness parameters $L_0, L_1$ for logistic regression problems with a nonconvex regularizer, which are the following problems

$$
\min_{x \in \mathbb{R}^d} \left\{ f(x) := \frac{1}{n} \sum_{i=1}^n f_i(x) := \frac{1}{n} \sum_{i=1}^n \underbrace{\log(1 + \exp(-b_i a_i^T x))}_{=:\tilde{f}_i(x)} + \lambda \underbrace{\sum_{j=1}^d \frac{x_j^2}{1 + x_j^2}}_{=:h(x)} \right\},
$$

where $a_i \in \mathbb{R}^d$ is the $i^{\text{th}}$ feature vector of matrix $A$ with its class label $b_i \in \{-1, 1\}$, $\lambda > 0$.

First, we can prove that $f(x)$ is $L$-smooth with $L = \frac{1}{4n}\|A\|^2 + 2\lambda$, and that each $f_i(x)$ is $\tilde{L}_i$-smooth with $\tilde{L}_i = \frac{1}{4}\|a_i\|^2 + 2\lambda$.

Next, we show that each $f_i(x)$ is generalized smooth with $L_0 = 2\lambda + \lambda\sqrt{d}\max_i \|a_i\|$ and $L_1 = \max_i \|a_i\|$, when the Hessian exists. By the fact that

$$
\nabla \tilde{f}_i(x) = -\frac{\exp(-b_i a_i^T x)}{1 + \exp(-b_i a_i^T x)} b_i a_i, \quad \text{and} \quad \nabla^2 \tilde{f}_i(x) = \frac{\exp(-b_i a_i^T x)}{(1 + \exp(-b_i a_i^T x))^2} b_i^2 a_i a_i^T,
$$

we have

$$
\begin{aligned}
\left\| \nabla^2 \tilde{f}_i(x) \right\| &\overset{b_i \in \{-1,1\}}{=} \frac{\exp(-b_i a_i^T x)}{(1 + \exp(-b_i a_i^T x))^2} \lambda_{\max}(a_i a_i^T) \\
&= \frac{\exp(-b_i a_i^T x)}{(1 + \exp(-b_i a_i^T x))^2} \|a_i\|^2 \\
&= \frac{\|a_i\|}{1 + \exp(-b_i a_i^T x)} \left\| \nabla \tilde{f}_i(x) \right\| \\
&\leq \|a_i\| \left\| \nabla \tilde{f}_i(x) \right\|.
\end{aligned}
\tag{16}
$$

After adding the nonconvex regularizer $h(x)$, we can show the following inequalities:

$$
\begin{aligned}
\left\|\nabla^2 f_i(x)\right\| &\leq \left\|\nabla^2 \tilde{f}_i(x)\right\| + \left\|\nabla^2 h(x)\right\| \\
&\leq \left\|\nabla^2 \tilde{f}_i(x)\right\| + 2\lambda,
\end{aligned} \tag{17}
$$

and

$$
\begin{aligned}
\|\nabla f_i(x)\| \geq \left\|\nabla \tilde{f}_i(x)\right\| - \|\nabla h(x)\| &= \left\|\nabla \tilde{f}_i(x)\right\| - \sqrt{\left(\frac{2\lambda x_1}{(1+x_1^2)^2}\right)^2 + \ldots + \left(\frac{2\lambda x_d}{(1+x_d^2)^2}\right)^2} \\
&\geq \left\|\nabla \tilde{f}_i(x)\right\| - \sqrt{\lambda^2 + \ldots + \lambda^2} \\
&= \left\|\nabla \tilde{f}_i(x)\right\| - \lambda\sqrt{d}.
\end{aligned} \tag{18}
$$

By combining inequalities (16), (17), and (18), we obtain

$$
\begin{aligned}
\left\|\nabla^2 f_i(x)\right\| &\leq \left\|\nabla^2 \tilde{f}_i(x)\right\| + 2\lambda \\
&\leq \|a_i\| \left\|\nabla \tilde{f}_i(x)\right\| + 2\lambda \\
&\leq 2\lambda + \lambda\sqrt{d} + \|a_i\| \|\nabla f_i(x)\|.
\end{aligned}
$$

In conclusion, $\left\|\nabla^2 f_i(x)\right\| \leq L_0 + L_1 \|\nabla f_i(x)\|$ with $L_0 \geq 2\lambda + \lambda\sqrt{d}$, and $L_1 \geq \|a_i\|$. This condition is equivalent to Assumption 3 (generalized smoothness) with $L_0, L_1$ (Chen et al., 2023, Theorem 1).

