# OpenReview forum: "Communication-efficient Algorithms Under Generalized Smoothness Assumptions"
_ICLR.cc/2025/Conference — Submitted to ICLR 2025_

### Official Review · Reviewer_LkH1 · 2024-11-01

**Soundness:** 3
**Presentation:** 3
**Contribution:** 2
**Rating:** 5
**Confidence:** 4

**Summary:**

This paper mainly considers the normalized EF21 algorithm in distributed settings. It generalizes the existing EF21 algorithm to the $(L_0,L_1)$-smoothness settings and achieves comparable convergence results. In the stochastic settings, the paper incorporates momentum into the algorithm to control the effect of gradient variance. Experimental results are generally consistent with the theoretical claims.

**Strengths:**

1. The paper is well-written and easy to follow. The notation definitions are clear.

2. The theorems are clear and complete. The theoretical results are novel and seem to be sound.

**Weaknesses:**

1. This is a technical solid paper, however, the technical novelty appears limited. I briefly went through the proof and my impression is that the proof primarily combines existing tools on $(L_0,L_1)$-smoothness, error feedback, and analysis of normalized gradient descent with momentum. These are well-established techniques, and the results, while solid, are not particularly surprising. It is possible that I overlooked something important, and I would appreciate it if you would point it out and allocate some space to briefly talk about the technical novelty in this case.

2. I am not sure why Theorem 2 implies the results of NSGD-M as the author mentioned in line 406. There is a mismatch that when $\sigma=0$, i.e. the deterministic case, the order of Theorem 2 is still $\mathcal{O}(T^{-1/4})$, while Theorem 1 in (Cutkosky \& Mehta, 2020) is $\mathcal{O}(T^{-1/2})$. I think maybe you should follow (Cutkosky \& Mehta, 2020) to allow a step size order shift regarding the magnitude of $\sigma$ to fix this.

[1] Ashok Cutkosky, and Harsh Mehta. "Momentum improves normalized sgd." International conference on machine learning. PMLR, 2020.

**Questions:**

1. In Figure 1, it seems that EF21 (without normalization) converges faster in the first place. Could you provide some explanations or comments for this, as it seems this figure may not be strong enough to prove the efficiency of normalized EF21?

2. I am curious whether clipped EF21 rather than normalized EF21 is also applicable to the $(L_0,L_1)$-smoothness setting in the paper.

---

> ### Author Response · Authors · 2024-11-19
>
> > **This is a technical solid paper, however, the technical novelty appears limited. I briefly went through the proof and my impression is that the proof primarily combines existing tools on $(L_0,L_1)$-smoothness, error feedback, and analysis of normalized gradient descent with momentum. These are well-established techniques, and the results, while solid, are not particularly surprising. It is possible that I overlooked something important, and I would appreciate it if you would point out and allocate some space to briefly talk about the technical novelty in this case.**
>
> Our paper addresses the open problem of studying communication-compression algorithms under generalized smoothness. Our convergence analysis introduces novel proof techniques that differ significantly from prior literature on communication-compression algorithms.
>
> **Unexplored Theoretical Understanding.**  Theoretical analysis of error feedback algorithms under generalized smoothness remains largely unexplored. Prior studies predominantly assume **traditional smoothness** to establish convergence guarantees for such algorithms.
>
> **Emerging Research on Generalized Smoothness.**  The analysis of gradient-based algorithms under generalized smoothness has gained recent attention only within the last five years. The **generalized smoothness condition** was first proposed by Zhang et al. (2020b) and later refined by Chen et al. (2023) to cover a broader range of machine learning problems. Very recent works, including Chen et al. (2023), Koloskova et al. (2023), and Hübler et al. (2024), have imposed generalized smoothness to analyze gradient-based algorithms.
>
> **The Role of Normalization in EF21 Under Generalized Smoothness.**  While the original EF21 designed for traditional $L$-smoothness can be extended to generalized $(L_0, L_1)$-smoothness, its performance is limited without normalization. Specifically:
>
> - Under generalized smoothness, deriving the convergence of EF21 can be done by proving $\Vert \nabla f(x) \Vert \leq B$, allowing the smoothness constant to be defined as $L := L_0 + L_1B$. However, this smoothness constant $L$ can be potentially small, resulting in slower convergence for EF21, as its step size is constrained to $\gamma = \mathcal{O}(1/L)$ to ensure theoretical convergence.
>
> - In contrast, normalized EF21 benefits from a step size of $\gamma = \gamma_0 / \sqrt{K+1}$, where $\gamma_0, K > 0$. This enables faster convergence, as demonstrated in our experiments for minimizing polynomial functions in Figure 1, and for solving non-convex logistic regression problems in Figure 2.
>
>
> **Novel Proof Techniques for EF21 under Generalized Smoothness.**  Our analysis demonstrates that normalized EF21 achieves a convergence rate under generalized smoothness equivalent to EF21 under traditional smoothness. However, our proof techniques differ from these previous works.
>
> **Lyapunov Function Innovation:**  We rely on different Lyapunov functions. For **EF21**, we use the Lyapunov function $V^k := f(x^k) - f^{\inf} + \frac{A}{n} \sum_{i=1}^n \Vert \nabla f_i(x^k) - v_i^k \Vert$, unlike Richtárik et al. (2021), which uses $V^k := f(x^k) - f^{\inf} + \frac{B}{n} \sum_{i=1}^n \Vert \nabla f_i(x^k) - v_i^k \Vert^2$.  For **EF21-SGDM**, we adopt $V^k := f(x^k) - f^{\inf} + \frac{C}{n} \sum_{i=1}^n \Vert v_i^k - g_i^k \Vert$, unlike Fatkhullin et al. (2024), which utilizes $V^k := f(x^k) - f^{\inf} + \frac{D}{n} \sum_{i=1}^n \Vert v_i^k - g_i^k \Vert^2 + \frac{E}{n} \sum_{i=1}^n \Vert g_i^k - \nabla f_i(x^k) \Vert^2$.
>
> **Convergence Rate Derivation:** These novel Lyapunov functions necessitate the development of new techniques to derive convergence rates that match those presented in the prior works. We employ **Lemma 2** to handle generalized smoothness. To obtain convergence rates for EF21 and EF21-SGDM, we use **Lemmas 4 and 9**, respectively, which align with the rates achieved by Richtárik et al. (2021) and Fatkhullin et al. (2024).
>
> Our approach not only builds on existing research but also introduces novel analysis techniques to expand the theoretical understanding of communication-compression algorithms under generalized smoothness.

---

> > ### Author Response · Authors · 2024-11-19
> >
> > > **I am not sure why Theorem 2 implies the results of NSGD-M as the author mentioned in line 406. There is a mismatch that when $\sigma=0$, i.e. the deterministic case, the order of Theorem 2 is still $\mathcal{O}(T^{-1/4})$, while Theorem 1 in (Cutkosky & Mehta, 2020) is $\mathcal{O}(T^{-1/2})$. I think maybe you should follow (Cutkosky & Mehta, 2020) to allow a step size order shift regarding the magnitude of $\sigma$ to fix this.**
> >
> > > **[1] Ashok Cutkosky, and Harsh Mehta. "Momentum improves normalized sgd." International conference on machine learning. PMLR, 2020.**
> >
> > We apologize for any lack of clarity in our previous discussion. Theorem 2 with $\alpha = 1$ provides the convergence bound for the distributed version of NSGD-M. To the best of our knowledge, the best convergence rate results for NSGD-M  under traditional smoothness and centralized regimes was initially obtained by Cutkosky & Mehta (2020).
> >
> > We respectfully disagree with your comment, as the analysis of NSGD-M by Cutkosky & Mehta (2020) assumes traditional smoothness, which is a simpler setting compared to generalized smoothness. Both Hübler et al. (2024) and our work focus on generalized smoothness, which presents a more challenging setting for analysis. Furthermore, by setting $\alpha = 1$, $n = 1$, and $\delta^{\inf} = 0$ in Theorem 2, we obtain a convergence bound that matches the $\mathcal{O}(T^{-1/4})$ rate derived by Hübler et al. (2024, Corollary 3) under generalized smoothness, as we discussed in Line 412-418.

---

> ### Author Response · Authors · 2024-11-19
>
> > **In Figure 1, it seems that EF21 (without normalization) converges faster in the first place. Could you provide some explanations or comments for this, as it seems this figure may not be strong enough to prove the efficiency of normalized EF21?**
>
> We believe that Figure 1 supports the efficiency of normalized EF21, as it demonstrates that for minimizing polynomial functions, normalized EF21 allows for larger stepsizes and achieves a faster convergence rate compared to EF21. Specifically, to reach $\Vert  \nabla f(x)\Vert^2 = 10^{-4}$, normalized EF21 requires roughly half the number of iterations as EF21 for different sets of hyper-parameters.
>
> In the rightmost plot, EF21 initially converges faster than normalized EF21 due to the small stepsize $\gamma_0 / \sqrt{K+1}$ for normalized EF21 at the beginning (when $K = 16,000$). Despite this, normalized EF21 ultimately achieves higher solution accuracy than EF21.

---

> > ### Author Response · Authors · 2024-11-19
> >
> > > **I am curious whether clipped EF21 rather than normalized EF21 is also applicable to the $(L_0,L_1)$-smoothness setting in the paper.**
> >
> > Extending our analysis techniques for normalized EF21 to analyze clipped EF21 is an interesting yet possible direction.
> >
> > Following Koloskova et al. (2023), analyzing EF21 with clipping $\text{Clip}_C(g) = \min(1, C / \Vert g\Vert)$ requires deriving descent inequalities under two scenarios: (1) when clipping is activated ($\Vert g\Vert > C$), and (2) when clipping is deactivated ($\Vert g\Vert \leq C$). However, two technical challenges must be addressed:
> >
> > **Transition from clipping on to clipping off:** Proving that the number of iterations where clipping is active ($\Vert g\Vert > C$) is finite and will eventually end.
> >
> > **Combining descent inequalities:** Combining the descent inequalities for the scenarios when clipping is active and inactive to establish the convergence rate.

---

> > > ### Comment · Reviewer_LkH1 · 2024-11-22
> > >
> > > Thanks for the detailed reply. I appreciate the authors' effort in explaining the novelty of this work. However, as I noted in Weakness 2 part, I would like to kindly point out that the $\mathcal{O}(T^{1/4})$ rate for $\sigma=0$ implied in the bound is not satisfying regardless the assumption. I think this is natural as even if you use the generalized smoothness assumption, you should get a convergence rate of $\mathcal{O}(T^{1/2})$ when you directly analyze the deterministic case.
> > >
> > > But never mind, I think this is not an important issue. I would like to keep the score as is.

---

### Official Review · Reviewer_Uygz · 2024-11-04

**Soundness:** 3
**Presentation:** 3
**Contribution:** 2
**Rating:** 5
**Confidence:** 4

**Summary:**

This paper presents the first convergence proof for normalized error feedback algorithms under generalized smoothness, a condition more applicable to a range of machine learning problems than traditional smoothness assumptions. Existing analyses either focus on single-node setups or make unrealistic assumptions (e.g., data heterogeneity) for distributed settings. Here, the authors propose distributed error feedback algorithms with normalization, achieving an $\(O(1/\sqrt{K})\) $ convergence rate for nonconvex problems without requiring data heterogeneity. The approach also enables stepsize tuning independent of problem parameters. Empirical results show that normalized EF21, which allows larger stepsizes, outperforms EF21 across tasks like polynomial minimization, logistic regression, and ResNet-20 training.

**Strengths:**

1 The paper clearly articulates the limitations of previous approaches and their constraints in proof, while explicitly highlighting the improvements and increased generalizability introduced by the proposed method.

2 This paper presents the first convergence proof for normalized error feedback algorithms under generalized smoothness in a decentralized setting, filling an existing theoretical gap on this issue.

3 The proof appears to follow a standard framework for decentralized problems, making it fairly credible.

**Weaknesses:**

1 The algorithm itself does not present significant innovation, as it primarily combines the standard EF21 with a normalization technique.

**Questions:**

Could you explain the specific challenges encountered in the proof after changing the assumptions and incorporating the normalization step, and how these challenges were addressed?

---

> ### Author Response · Authors · 2024-11-19
>
> > **The algorithm itself does not present significant innovation, as it primarily combines the standard EF21 with a normalization technique.**
>
> > **Could you explain the specific challenges encountered in the proof after changing the assumptions and incorporating the normalization step, and how these challenges were addressed?**
>
> Our paper addresses the open problem of studying communication-compression algorithms under generalized smoothness. Our convergence analysis introduces novel proof techniques that differ significantly from prior literature on communication-compression algorithms.
>
> **Unexplored Theoretical Understanding.**  Theoretical analysis of error feedback algorithms under generalized smoothness remains largely unexplored. Prior studies predominantly assume **traditional smoothness** to establish convergence guarantees for such algorithms.
>
> **Emerging Research on Generalized Smoothness.**  The analysis of gradient-based algorithms under generalized smoothness has gained recent attention only within the last five years. The **generalized smoothness condition** was first proposed by Zhang et al. (2020b) and later refined by Chen et al. (2023) to cover a broader range of machine learning problems. Very recent works, including Chen et al. (2023), Koloskova et al. (2023), and Hübler et al. (2024), have imposed generalized smoothness to analyze gradient-based algorithms.
>
> **The Role of Normalization in EF21 Under Generalized Smoothness.**  While the original EF21 designed for traditional $L$-smoothness can be extended to generalized $(L_0, L_1)$-smoothness, its performance is limited without normalization. Specifically:
>
> - Under generalized smoothness, deriving the convergence of EF21 can be done by proving $\Vert \nabla f(x) \Vert \leq B$, allowing the smoothness constant to be defined as $L := L_0 + L_1B$. However, this smoothness constant $L$ can be potentially small, resulting in slower convergence for EF21, as its step size is constrained to $\gamma = \mathcal{O}(1/L)$ to ensure theoretical convergence.
>
> - In contrast, normalized EF21 benefits from a step size of $\gamma = \gamma_0 / \sqrt{K+1}$, where $\gamma_0, K > 0$. This enables faster convergence, as demonstrated in our experiments for minimizing polynomial functions in Figure 1, and for solving non-convex logistic regression problems in Figure 2.
>
>
> **Novel Proof Techniques for EF21 under Generalized Smoothness.**  Our analysis demonstrates that normalized EF21 achieves a convergence rate under generalized smoothness equivalent to EF21 under traditional smoothness. However, our proof techniques differ from these previous works.
>
> **Lyapunov Function Innovation:**  We rely on different Lyapunov functions. For **EF21**, we use the Lyapunov function $V^k := f(x^k) - f^{\inf} + \frac{A}{n} \sum_{i=1}^n \Vert \nabla f_i(x^k) - v_i^k \Vert$, unlike Richtárik et al. (2021), which uses $V^k := f(x^k) - f^{\inf} + \frac{B}{n} \sum_{i=1}^n \Vert \nabla f_i(x^k) - v_i^k \Vert^2$.  For **EF21-SGDM**, we adopt $V^k := f(x^k) - f^{\inf} + \frac{C}{n} \sum_{i=1}^n \Vert v_i^k - g_i^k \Vert$, unlike Fatkhullin et al. (2024), which utilizes $V^k := f(x^k) - f^{\inf} + \frac{D}{n} \sum_{i=1}^n \Vert v_i^k - g_i^k \Vert^2 + \frac{E}{n} \sum_{i=1}^n \Vert g_i^k - \nabla f_i(x^k) \Vert^2$.
>
> **Convergence Rate Derivation:** These novel Lyapunov functions necessitate the development of new techniques to derive convergence rates that match those presented in the prior works. We employ **Lemma 2** to handle generalized smoothness. To obtain convergence rates for EF21 and EF21-SGDM, we use **Lemmas 4 and 9**, respectively, which align with the rates achieved by Richtárik et al. (2021) and Fatkhullin et al. (2024).
>
> Our approach not only builds on existing research but also introduces novel analysis techniques to expand the theoretical understanding of communication-compression algorithms under generalized smoothness.

---

> > ### Comment · Reviewer_Uygz · 2024-11-27
> >
> > Thank you for your response. I will maintain my score.

---

### Official Review · Reviewer_5w1G · 2024-11-06

**Soundness:** 3
**Presentation:** 3
**Contribution:** 3
**Rating:** 5
**Confidence:** 4

**Summary:**

**Summary.** The authors present the convergence guarantees for normalized error feedback algorithms for minimizing objectives satisfying generalized smoothness assumption. The proposed algorithm achieves $O(1/\sqrt{K})$ convergence rate for minimizing nonconvex problems. The authors also extend the analysis to stochastic settings. Finally, the authors show via numerical experiments that normalized EF21 outperforms EF21 on various tasks.

**Strengths:**

**Strengths.** Here, I list the major strengths of the paper.

- The authors present the first analysis of normalized error feedback algorithms in a distributed setting for minimizing non-convex objectives satisfying generalized smoothness conditions. The proposed algorithm matches the guarantees of the standard error feedback algorithms under classical smoothness assumptions.

-  The authors also show convergence under the stochastic settings.
- The authors numerically evaluate the performance of the proposed algorithm.

**Weaknesses:**

**Weaknesses.** Here, I list my major concerns and comments.

- The discussion on the requirement of heterogeneity assumption by the earlier algorithms and the proposed algorithm not requiring the assumption is slightly misleading. In my understanding, heterogeneity assumption is usually needed if the local machines conduct multiple updates while the proposed algorithm, Normalized EF21, only conducts a single local update. So naturally the heterogeneity assumption is not required in contrast to the works of Crawshaw et al. (2024) and Liu et al. (2022) where the local machines conduct multiple local updates.  The authors should clarify the connection between their single local update approach and the lack of heterogeneity assumptions. Additionally, how does the proposed algorithm perform with multiple local updates, and whether heterogeneity assumptions would then become necessary?

- Why both Assumptions 1 and 2 are required? Assumption 2 implies Assumption 1, moreover, I believe it should be possible to conduct analysis only with Assumption 1.  Please justify the inclusion of both assumptions or revise the analysis to rely only on Assumption 1 if possible.

- For the stochastic setting, usually, the speed-up with the number of clients in the performance is desired. However, the presented analysis does not capture the effect of the number of clients on the performance.  Does the proposed algorithm achieve linear speed-up with the number of clients? Please analyze or discuss the scalability of the proposed algorithm with respect to the number of clients.

- In Theorem 1, is there an upper bound on the requirement of $\gamma_0$? Why such a bound is not necessary for Theorem 1 but is required in Theorem 2?

- The presented numerical results are very weak. The authors have compared the performance of their algorithm only against the EF21 algorithm. However, there are several federated baselines (with and without compression) against whom the authors should compare their algorithm. For example, the baselines like FedAvg, CHOCO-SGD, Momentum based algorithms, etc.

**Questions:**

Please see the weaknesses above.

---

> ### Author Response · Authors · 2024-11-19
>
> > **The discussion on the requirement of heterogeneity assumption by the earlier algorithms and the proposed algorithm not requiring the assumption is slightly misleading. In my understanding, heterogeneity assumption is usually needed if the local machines conduct multiple updates while the proposed algorithm, Normalized EF21, only conducts a single local update. So naturally the heterogeneity assumption is not required in contrast to the works of Crawshaw et al. (2024) and Liu et al. (2022) where the local machines conduct multiple local updates. The authors should clarify the connection between their single local update approach and the lack of heterogeneity assumptions. Additionally, how does the proposed algorithm perform with multiple local updates, and whether heterogeneity assumptions would then become necessary?**
>
> **Relaxing the Uniformly Bounded Data Heterogeneity Assumption.** Our results for normalized EF21 in both deterministic and stochastic distributed optimization do not rely on the uniformly bounded data heterogeneity assumption, unlike the works of Crawshaw et al. (2024) and Liu et al. (2022). Instead, our convergence bounds in Theorems 1 and 2 include an additive constant related to $\delta^{\inf} := f^{\inf} - \frac{1}{n} \sum_i f_i^{\inf}$, which quantifies data heterogeneity.
>
> **Uniformly Bounded Data Heterogeneity: Pessimistic but Avoidable.** The uniformly bounded data heterogeneity assumption is often used to analyze federated learning algorithms, both under traditional smoothness and generalized smoothness, as analyzed by Crawshaw et al. (2024) and Liu et al. (2022). However, this assumption has been shown by [1] to be overly pessimistic for modeling real-world data heterogeneity. It is possible to rather analyze federated learning algorithms without imposing these uniformly bounded data heterogeneity under traditional smoothness, such as FedAvg (or Local SGD) by [2], and FedProx by [3]. We believe an interesting and open problem remains. That is, developing federated algorithms with compressed communication under generalized smoothness, and analyzing their convergence without relying on the uniformly bounded heterogeneity assumption.
>
> [1] Wang, Jianyu, et al. "On the unreasonable effectiveness of federated averaging with heterogeneous data." arXiv preprint arXiv:2206.04723 (2022).
>
> [2] Gorbunov, Eduard, Filip Hanzely, and Peter Richtárik. "Local sgd: Unified theory and new efficient methods." International Conference on Artificial Intelligence and Statistics. PMLR, 2021.
>
> [3] Yuan, Xiaotong, and Ping Li. "On convergence of FedProx: Local dissimilarity invariant bounds, non-smoothness and beyond." Advances in Neural Information Processing Systems 35 (2022): 10752-10765.

---

> ### Author Response · Authors · 2024-11-19
>
> > **Why both Assumptions 1 and 2 are required? Assumption 2 implies Assumption 1, moreover, I believe it should be possible to conduct analysis only with Assumption 1. Please justify the inclusion of both assumptions or revise the analysis to rely only on Assumption 1 if possible.**
>
> Assumptions 1 and 2 are standard for deriving distributed optimization, satisfied by many machine learning problems, and imposed to model data heterogeneity.
>
> - **Assumptions 1 and 2 are standard for distributed optimization:** Assumptions 1 and 2 are standard for deriving distributed stochastic gradient descent for nonconvex, $L$-smooth problems, e.g., [1].
>
> - **Assumptions 1 and 2 are satisfied by many machine learning problems:** Many machine learning problems can be cast into optimization problems with non-negative objective functions, such as squared error loss in linear regression, logistic loss in logistic regression, and cross-entropy loss in neural network training.  If $f(x)$ and each $f_i(x)$ are non-negative, then Assumptions 1 and 2 imply $f^{\inf} = f_i^{\inf} = 0$. This leads to our convergence bounds without an additive constant related to data heterogeneity, i.e., $\delta^{\inf} = 0$.
>
> - **Assumptions 1 and 2 model data heterogeneity:** Assumptions 1 and 2 are required to model data heterogeneity. Instead of imposing a uniformly bounded data heterogeneity condition, our convergence bounds include an additive constant related to $\delta^{\inf} := f^{\inf} - \frac{1}{n} \sum_i f_i^{\inf}$, which quantifies data heterogeneity.
>
> [1] Khaled, Ahmed and Peter Richtárik. “Better Theory for SGD in the Nonconvex World.” In Transactions on Machine Learning Research, 2023. P. 2835-8856.

---

> > ### Author Response · Authors · 2024-11-19
> >
> > > **For the stochastic setting, usually, the speed-up with the number of clients in the performance is desired. However, the presented analysis does not capture the effect of the number of clients on the performance. Does the proposed algorithm achieve linear speed-up with the number of clients? Please analyze or discuss the scalability of the proposed algorithm with respect to the number of clients.**
> >
> > The existing literature on error feedback methods, such as EF14 and EF21, primarily focuses on the class of biased compressors applied to gradients. These methods use gradients modified by the compressors to update the iterates. Consequently, their convergence results do not demonstrate a speed-up with an increasing number of clients, which remains an open problem. A recent effort addressing this issue is presented in [1], where the iteration complexity of error feedback methods is refined using a data sparsity measure $c$.
> >
> > From Theorem 2, which establishes the convergence of EF21-SGDM under generalized smoothness, the derived convergence bound comprises two terms. The first term is independent of the compression parameter $\alpha$ and features a $\sigma / \sqrt{n}$-term for the gradient norm, similar to the $\sigma^2 / n$-term for the squared gradient norm shown in Theorem 3 of Fatkhullin et al. (2024). The second term, dependent on $\alpha$, includes a $\sigma$-term but lacks the speed-up with the number of clients $n$. This limitation arises because EF21-SGDM is a generalized version of EF21, and the theoretical challenge regarding speed-up in EF21 also extends to EF21-SGDM and other variants derived from EF21.
> >
> > [1] Richtarik, Peter, Elnur Gasanov, and Konstantin Burlachenko. "Error Feedback Shines when Features are Rare." arXiv preprint arXiv:2305.15264 (2023).

---

> > > ### Author Response · Authors · 2024-11-19
> > >
> > > > **In Theorem 1, is there an upper bound on the requirement of $\gamma_0$? Why such a bound is not necessary for Theorem 1 but is required in Theorem 2?**
> > >
> > > Unlike Theorem 1 that applies for any $\gamma_0>0$, Theorem 2 imposes an upper bound on $\gamma_0$ to satisfy the condition $\frac{\gamma^2}{\eta a_3 (K+1)} \leq \frac{1}{2}$ in Lemma 9, which is necessary to ensure the convergence for normalized EF21-SGDM. We believe that it may be possible to relax the requirement for an upper bound on $\gamma_0$ in Theorem 2. However, doing so would likely lead to more complex derivation steps for obtaining convergence results.

---

> > > > ### Author Response · Authors · 2024-11-19
> > > >
> > > > > **The presented numerical results are very weak. The authors have compared the performance of their algorithm only against the EF21 algorithm. However, there are several federated baselines (with and without compression) against whom the authors should compare their algorithm. For example, the baselines like FedAvg, CHOCO-SGD, Momentum based algorithms, etc.**
> > > >
> > > > We believe this concern is minor, as also suggested by Reviewer nRsr.
> > > >
> > > > - **Empirical Comparisons Between Normalized EF21 and EF21.**  We focus our comparisons on normalized EF21 and EF21, because these algorithms are most relevant to our research scope. Our empirical results demonstrate that, for solving generalized smooth problems, incorporating normalization into EF21 is essential to enable larger step sizes and achieve faster convergence rates.
> > > >
> > > > - **Comparisons Against Other Algorithms.**  Comparisons between normalized EF21 and other federated (e.g., FedAvg) or decentralized algorithms (e.g., CHOCO-SGD) fall outside the scope of our study. This is because normalized EF21 performs a single local update, unlike FedAvg, and also requires a central server to maintain the global solution, unlike CHOCO-SGD. However, we believe that extending normalized EF21 for federated and decentralized optimization is an important future research direction. For instance, it remains an open challenge to design and analyze federated algorithms with compressed communication under generalized smoothness without relying on additional restrictive assumptions, such as uniformly bounded heterogeneity (as discussed by Crawshaw et al., 2024, and Liu et al., 2022).

---

### Official Review · Reviewer_nRsr · 2024-11-08

**Soundness:** 3
**Presentation:** 2
**Contribution:** 2
**Rating:** 5
**Confidence:** 3

**Summary:**

This paper proposes normalized error feedback algorithms across a variety of settings and provides convergence guarantees for this family of algorithms. The paper also provides experimental results on various settings such as logistic regression with non-convex regularizer. Notably, the convergence guarantees only rely on the general smoothness condition, which is a generalization of the classical smoothness condition.

**Strengths:**

The paper relaxes the classical smoothness condition to the generalized smoothness condition. Furthermore, the convergence guarantee is competitive and matches the SOTA.

**Weaknesses:**

My main concern with this paper is the only notable contribution is the relaxation from the classical smoothness condition to the generalized smoothness condition. While it is indeed an interesting result, the analysis is known from Chen et al. 2023 and various other works. And so, the theoretical analysis is not too impressive.

The empirical settings are a bit limited and not well-motivated but this is only a minor point.

The paper does not present well the communication aspect of the algorithm, nor its proof idea.

Minor comment:
- The word "expectation" should be capitalized in Table 1.

**Questions:**

- The paper does not present the communication-efficient aspect of the algorithm well, and it is not clear what is the communication protocol. The paper should state this explicitly. Furthermore, there is no analysis on the communication complexity. Is it the same as the sample complexity?

---

> ### Author Response · Authors · 2024-11-19
>
> > **My main concern with this paper is the only notable contribution is the relaxation from the classical smoothness condition to the generalized smoothness condition. While it is indeed an interesting result, the analysis is known from Chen et al. 2023 and various other works. And so, the theoretical analysis is not too impressive.**
>
> Our paper addresses the open problem of studying communication-compression algorithms under generalized smoothness. Our convergence analysis introduces novel proof techniques that differ significantly from prior literature on communication-compression algorithms.
>
> **Unexplored Theoretical Understanding.**  Theoretical analysis of error feedback algorithms under generalized smoothness remains largely unexplored. Prior studies predominantly assume **traditional smoothness** to establish convergence guarantees for such algorithms.
>
> **Emerging Research on Generalized Smoothness.**  The analysis of gradient-based algorithms under generalized smoothness has gained recent attention only within the last five years. The **generalized smoothness condition** was first proposed by Zhang et al. (2020b) and later refined by Chen et al. (2023) to cover a broader range of machine learning problems. Very recent works, including Chen et al. (2023), Koloskova et al. (2023), and Hübler et al. (2024), have imposed generalized smoothness to analyze gradient-based algorithms.
>
> **The Role of Normalization in EF21 Under Generalized Smoothness.**  While the original EF21 designed for traditional $L$-smoothness can be extended to generalized $(L_0, L_1)$-smoothness, its performance is limited without normalization. Specifically:
>
> - Under generalized smoothness, deriving the convergence of EF21 can be done by proving $\Vert \nabla f(x) \Vert \leq B$, allowing the smoothness constant to be defined as $L := L_0 + L_1B$. However, this smoothness constant $L$ can be potentially small, resulting in slower convergence for EF21, as its step size is constrained to $\gamma = \mathcal{O}(1/L)$ to ensure theoretical convergence.
>
> - In contrast, normalized EF21 benefits from a step size of $\gamma = \gamma_0 / \sqrt{K+1}$, where $\gamma_0, K > 0$. This enables faster convergence, as demonstrated in our experiments for minimizing polynomial functions in Figure 1, and for solving non-convex logistic regression problems in Figure 2.
>
>
> **Novel Proof Techniques for EF21 under Generalized Smoothness.**  Our analysis demonstrates that normalized EF21 achieves a convergence rate under generalized smoothness equivalent to EF21 under traditional smoothness. However, our proof techniques differ from these previous works.
>
> **Lyapunov Function Innovation:**  We rely on different Lyapunov functions. For **EF21**, we use the Lyapunov function $V^k := f(x^k) - f^{\inf} + \frac{A}{n} \sum_{i=1}^n \Vert \nabla f_i(x^k) - v_i^k \Vert$, unlike Richtárik et al. (2021), which uses $V^k := f(x^k) - f^{\inf} + \frac{B}{n} \sum_{i=1}^n \Vert \nabla f_i(x^k) - v_i^k \Vert^2$.  For **EF21-SGDM**, we adopt $V^k := f(x^k) - f^{\inf} + \frac{C}{n} \sum_{i=1}^n \Vert v_i^k - g_i^k \Vert$, unlike Fatkhullin et al. (2024), which utilizes $V^k := f(x^k) - f^{\inf} + \frac{D}{n} \sum_{i=1}^n \Vert v_i^k - g_i^k \Vert^2 + \frac{E}{n} \sum_{i=1}^n \Vert g_i^k - \nabla f_i(x^k) \Vert^2$.
>
> **Convergence Rate Derivation:** These novel Lyapunov functions necessitate the development of new techniques to derive convergence rates that match those presented in the prior works. We employ **Lemma 2** to handle generalized smoothness. To obtain convergence rates for EF21 and EF21-SGDM, we use **Lemmas 4 and 9**, respectively, which align with the rates achieved by Richtárik et al. (2021) and Fatkhullin et al. (2024).
>
> Our approach not only builds on existing research but also introduces novel analysis techniques to expand the theoretical understanding of communication-compression algorithms under generalized smoothness.

---

> > ### Author Response · Authors · 2024-11-19
> >
> > > **The empirical settings are a bit limited and not well-motivated but this is only a minor point.**
> >
> >  We believe this concern is minor, as you suggest.
> >
> > - **Empirical Comparisons Between Normalized EF21 and EF21.**  We focus our comparisons on normalized EF21 and EF21, because these algorithms are most relevant to our research scope. Our empirical results demonstrate that, for solving generalized smooth problems, incorporating normalization into EF21 is essential to enable larger step sizes and achieve faster convergence rates.
> >
> > - **Comparisons Against Other Algorithms.**  Comparisons between normalized EF21 and other federated (e.g., FedAvg) or decentralized algorithms (e.g., CHOCO-SGD) fall outside the scope of our study. This is because normalized EF21 performs a single local update, unlike FedAvg, and also requires a central server to maintain the global solution, unlike CHOCO-SGD. However, we believe that extending normalized EF21 for federated and decentralized optimization is an important future research direction. For instance, it remains an open challenge to design and analyze federated algorithms with compressed communication under generalized smoothness without relying on additional restrictive assumptions, such as uniformly bounded heterogeneity (as discussed by Crawshaw et al., 2024, and Liu et al., 2022).

---

> > > ### Author Response · Authors · 2024-11-19
> > >
> > > > **The paper does not present the communication-efficient aspect of the algorithm well, and it is not clear what is the communication protocol. The paper should state this explicitly. Furthermore, there is no analysis on the communication complexity. Is it the same as the sample complexity?**
> > >
> > > We apologize for not discussing in detail the communication-efficient aspects and proof ideas of normalized EF21 and normalized EF21-SGDM. Below, we provide a detailed explanation:
> > >
> > > **Downlink Communication Compression.**  Normalized EF21 and normalized EF21-SGDM reduce communication costs from clients to the server by transmitting only the compressed vector $\Delta_i^k$ from each client $i$ to the central server.
> > > At each client, $\Delta_i^k$ is used to update the memory $v_i^k$.
> > >
> > > At the server, $\Delta_i^k$ is aggregated as $\frac{1}{n} \sum_{i=1}^n \Delta_i^k$, which is used to update $v^{k+1} = v^k + \frac{1}{n} \sum_{i=1}^n \Delta_i^k$. This process enables efficient updates of the next iterates $x^{k+1}$.
> > >
> > > **Proof sketch.**  Our analysis builds on the approaches for EF21 by Richtárik et al. (2021) and EF21-SGDM by Fatkhullin et al. (2024). However, we employ distinct Lyapunov functions and novel techniques to achieve matching convergence rates under generalized smoothness.
> > >
> > > - For **Normalized EF21**, we establish the descent inequality using the Lyapunov function  $V^k := f(x^k) - f^{\inf} + \frac{A}{n} \sum_{i=1}^n \Vert \nabla f_i(x^k) - v_i^k \Vert$  as shown in Lemma 5. Next, we utilize Lemma 2 to handle generalized smoothness and Lemma 4 to derive the convergence rate of $\mathcal{O}(1/\sqrt{K})$. Unlike EF21 (Richtárik et al., 2021), normalized EF21 achieves step sizes independent of Lipschitz smoothness parameters, offering greater flexibility.
> > >
> > > - For **Normalized EF21-SGDM**,  We establish the descent inequality using $V^k := f(x^k) - f^{\inf} + \frac{C}{n} \sum_{i=1}^n \Vert v_i^k - g_i^k \Vert$  as shown in Lemmas 6, 7, and 8. Next, we derive the convergence rate of $\mathcal{O}(1/K^{1/4})$ using Lemma 9.  Here, normalized EF21-SGDM requires $\gamma = \mathcal{O}(1/K^{1/4})$, whereas EF21-SGDM requires $\gamma = \mathcal{O}(1/K^{1/2})$.
> > >
> > > - Furthermore, the communication complexity differs from the iteration complexity by a constant factor. The communication complexity is determined by multiplying the iteration complexity by the number of bits communicated per iteration, which remains constant.

---

> > > > ### Author Response · Authors · 2024-11-19
> > > >
> > > > > **The word "expectation" should be capitalized in Table 1.**
> > > >
> > > > Thank you for pointing this typo out. We will fix this in the revised manuscript.

---

### Author Response · Authors · 2024-11-19

We thank all reviewers for appreciating our contributions on providing the first convergence analysis for normalized error feedback algorithms for distributed optimization under generalized smoothness. In particular, Reviewer nRsr, 5w1G, and Uygz highlights that our novel convergence results of normalized error feedback algorithms under $(L_0,L_1)$-smoothness match prior results of standard error feedback algorithms under traditional smoothness.

Next, we would like to address two main concerns shared among the reviewers.

> **Concern 1) Convergence results rely on the analyses known from prior works, e.g. Chen et al. 2023. Technical novelty appears limited, because technical proof relies on well-established techniques on  $(L_0,L_1)$-smoothness, error feedback, and analysis of normalized gradient descent with momentum.**

Our paper addresses the open problem of studying communication-compression algorithms under generalized smoothness. Our convergence analysis introduces novel proof techniques that differ significantly from prior literature on communication-compression algorithms.

**Unexplored Theoretical Understanding.**  Theoretical analysis of error feedback algorithms under generalized smoothness remains largely unexplored. Prior studies predominantly assume **traditional smoothness** to establish convergence guarantees for such algorithms.

**Emerging Research on Generalized Smoothness.**  The analysis of gradient-based algorithms under generalized smoothness has gained recent attention only within the last five years. The **generalized smoothness condition** was first proposed by Zhang et al. (2020b) and later refined by Chen et al. (2023) to cover a broader range of machine learning problems. Very recent works, including Chen et al. (2023), Koloskova et al. (2023), and Hübler et al. (2024), have imposed generalized smoothness to analyze gradient-based algorithms.

**The Role of Normalization in EF21 Under Generalized Smoothness.**  While the original EF21 designed for traditional $L$-smoothness can be extended to generalized $(L_0, L_1)$-smoothness, its performance is limited without normalization. Specifically:

- Under generalized smoothness, deriving the convergence of EF21 can be done by proving $\Vert \nabla f(x) \Vert \leq B$, allowing the smoothness constant to be defined as $L := L_0 + L_1B$. However, this smoothness constant $L$ can be potentially small, resulting in slower convergence for EF21, as its step size is constrained to $\gamma = \mathcal{O}(1/L)$ to ensure theoretical convergence.

- In contrast, normalized EF21 benefits from a step size of $\gamma = \gamma_0 / \sqrt{K+1}$, where $\gamma_0, K > 0$. This enables faster convergence, as demonstrated in our experiments for minimizing polynomial functions in Figure 1, and for solving non-convex logistic regression problems in Figure 2.


**Novel Proof Techniques for EF21 under Generalized Smoothness.**  Our analysis demonstrates that normalized EF21 achieves a convergence rate under generalized smoothness equivalent to EF21 under traditional smoothness. However, our proof techniques differ from these previous works.

**Lyapunov Function Innovation:**  We rely on different Lyapunov functions. For **EF21**, we use the Lyapunov function $V^k := f(x^k) - f^{\inf} + \frac{A}{n} \sum_{i=1}^n \Vert \nabla f_i(x^k) - v_i^k \Vert$, unlike Richtárik et al. (2021), which uses $V^k := f(x^k) - f^{\inf} + \frac{B}{n} \sum_{i=1}^n \Vert \nabla f_i(x^k) - v_i^k \Vert^2$.  For **EF21-SGDM**, we adopt $V^k := f(x^k) - f^{\inf} + \frac{C}{n} \sum_{i=1}^n \Vert v_i^k - g_i^k \Vert$, unlike Fatkhullin et al. (2024), which utilizes $V^k := f(x^k) - f^{\inf} + \frac{D}{n} \sum_{i=1}^n \Vert v_i^k - g_i^k \Vert^2 + \frac{E}{n} \sum_{i=1}^n \Vert g_i^k - \nabla f_i(x^k) \Vert^2$.

**Convergence Rate Derivation:** These novel Lyapunov functions necessitate the development of new techniques to derive convergence rates that match those presented in the prior works. We employ **Lemma 2** to handle generalized smoothness. To obtain convergence rates for EF21 and EF21-SGDM, we use **Lemmas 4 and 9**, respectively, which align with the rates achieved by Richtárik et al. (2021) and Fatkhullin et al. (2024).

Our approach not only builds on existing research but also introduces novel analysis techniques to expand the theoretical understanding of communication-compression algorithms under generalized smoothness.

---

> ### Author Response · Authors · 2024-11-19
>
> > **Concern 2) Empirical results only compare normalized EF21 against EF21, not against other federated algorithms, such as FedAvg, CHOCO-SGD, Momentum-based algorithms, etc.**
>
> We believe this concern is minor, as also suggested by Reviewer nRsr.
>
> **Empirical Comparisons Between Normalized EF21 and EF21.**  We focus our comparisons on normalized EF21 and EF21, because these algorithms are most relevant to our research scope. Our empirical results demonstrate that, for solving generalized smooth problems, incorporating normalization into EF21 is essential to enable larger step sizes and achieve faster convergence rates.
>
> **Comparisons Against Other Algorithms.**  Comparisons between normalized EF21 and other federated (e.g., FedAvg) or decentralized algorithms (e.g., CHOCO-SGD) fall outside the scope of our study. This is because normalized EF21 performs a single local update, unlike FedAvg, and also requires a central server to maintain the global solution, unlike CHOCO-SGD. However, we believe that extending normalized EF21 for federated and decentralized optimization is an important future research direction. For instance, it remains an open challenge to design and analyze federated algorithms with compressed communication under generalized smoothness without relying on additional restrictive assumptions, such as uniformly bounded heterogeneity (as discussed by Crawshaw et al., 2024, and Liu et al., 2022).

---

### Meta-Review · Area_Chair_Vqi5 · 2024-12-20

**Metareview:**

The paper presents a convergence proof for normalized error feedback algorithms under generalized smoothness. Existing analysis either focus on single-node setups or make unrealistic assumptions (e.g., data heterogeneity) for distributed settings. The authors propose a distributed error feedback algorithms with normalization, achieving an $(O(1/\sqrt{K})) $ convergence rate for nonconvex problems without requiring data heterogeneity. All the reviewers agreed that the paper has merits, nevertheless none of them was enthusiastic enough to champion the paper to get accepted.

**Additional Comments On Reviewer Discussion:**

The rebuttal did not change the opinion of the reviewers.

---

### Decision · Program_Chairs · 2025-01-22

Reject